



# Improving predictions of land-atmosphere interactions based on a hybrid data assimilation and machine learning method

Xinlei He[1], Yanping Li[2], Shaomin Liu[1*], Tongren Xu[1], Fei Chen[3], Zhenhua Li[2], Zhe Zhang[2], Rui Liu[4], Lisheng Song[5], Ziwei Xu[1], Zhixing Peng[1], Chen Zheng[6]

[1]State Key Laboratory of Earth Surface Processes and Resource Ecology, School of Natural Resources, Faculty of Geographical Science, Beijing Normal University, Beijing, China
[2]School of Environment and Sustainability, University of Saskatchewan, Saskatoon, SK, Canada
[3]National Center for Atmospheric Research, Boulder, CO, USA
[4]Institute of Urban Study, School of Environmental and Geographical Sciences (SEGS), Shanghai Normal University, Shanghai, China
[5]School of Geography and Tourism, Anhui Normal University, Wuhu, China
[6]Institute of Geographic Sciences and Natural Resources Research, Chinese Academy of Sciences, Beijing, China

*Correspondence to*: Shaomin Liu (smliu@bnu.edu.cn)

**Abstract.** The energy and moisture exchange between the land surface and atmospheric boundary layer plays a critical role in regional climate simulations. This paper implemented a hybrid data assimilation and machine learning framework (DA-ML method) into the Weather Research and Forecasting (WRF) model to optimize surface soil and vegetation conditions. The hybrid method can integrate remotely sensed leaf area index (LAI), multi-source soil moisture (SM) observations, and land surface models (LSMs) to accurately describe land surface states and fluxes. The performance of the hybrid method on the regional climate was evaluated in the Heihe River Basin (HRB), the second largest endorheic river basin in Northwest China. The findings indicate that the DA-ML method improved the estimation of evapotranspiration (ET) and generated a spatial distribution consistent with the ML-based watershed ET (ETMap). The WRF simulations overestimated (underestimated) the air temperature (specific humidity) in the vegetated areas of the HRB. In contrast, the estimated air temperature and specific humidity from WRF (DA-ML) agree well with the observations, especially in the midstream oasis. The DA-ML framework enhanced oasis-desert interactions by improving the soil and vegetation characteristics. The wetting and cooling effects and wind shield effects of the oasis were enhanced by the DA-ML. The wetting and cooling effect of the oasis can transfer water vapor to the surrounding desert, which benefits the oasis-desert ecosystem. The results show that the wetting and cooling effects only negligibly changed the local precipitation in the midstream oasis. However, upstream of the HRB, the integration of LAI and SM will induce water vapor intensification and promote precipitation, particularly on windward slopes.



# 1 Introduction

Land-atmosphere interactions are an essential component of the hydrological cycle and factors that influence climate change (Nelli et al., 2020; Zhou et al., 2022). Terrestrial components, such as soil and vegetation, play a crucial role in atmospheric processes, such as changes in evapotranspiration (ET), which affect the water vapor content in the atmosphere (Gentine et al., 2019; Sawada et al., 2015). Soil and vegetation processes directly affect surface water vapor transport and energy circulation, particularly in the arid vegetated area (Erlandsen et al., 2017; Gao et al., 2008; Liu et al., 2018a; Zhang et al., 2017a, 2019; Zhao et al., 2021). Such surface variability affects the available energy distribution at the land surface and has additional effects on the sensible and latent heat fluxes, surface temperature, and water vapor (Sawada et al., 2015; Wen et al., 2012). Although land surface models (LSMs) have been improved incrementally in the past decades, it is still challenging to effectively couple LSMs with atmospheric models to improve the description of land-atmosphere interactions (Chen and Dudhia, 2001; Liu et al., 2021). The success of the coupling depends not only on the sophisticated physical processes of the LSMs, but also on soil and vegetation characteristics and the accurate characterization of the water vapor fluxes at the land-atmosphere interface (Gentine et al., 2019; Zhang et al., 2021b, 2022).

The development of earth observation technology has provided important opportunities to study land-atmosphere interactions using the data assimilation (DA) method (Liang et al., 2021). The Land Data Assimilation System (LDAS) has been widely developed and applied in recent years under various hydrological and vegetation conditions (Wu et al., 2022; Xia et al., 2019). It uses remotely sensed observations to constrain model physical processes and empirical parameters to improve water-energy-carbon flux simulations (Tian et al., 2022; Zhao and Yang, 2018). A series of studies have assimilated satellite-retrieved leaf area index (LAI), soil moisture (SM), and microwave brightness temperature observations into LSMs and improved simulations of ET, runoff, and gross primary productivity (GPP) (Ahmad et al., 2022; He et al., 2021; Ling et al., 2019; Seo et al., 2021; Xie et al., 2017; Xu et al., 2021). In addition, DA can improve the initial conditions of regional climate models (RCMs) and enhance the capability of the models in simulating land-atmosphere interactions (Pan et al., 2017; Yi et al., 2021). Several studies have also shown that the assimilation of surface pressure, air temperature, humidity, wind speed, and lightning observations into the Weather Research and Forecasting (WRF) model can improve the simulation of atmospheric state variables and the accuracy of weather prediction (Campo et al., 2009; Cazes Boezio and Ortelli, 2019; Comellas Prat et al., 2021; Grzeschik et al., 2008; Pilguj et al., 2019).

Machine learning (ML) algorithms have been increasingly applied in earth and environmental modeling studies to predict land surface variables at various spatial and temporal scales (Jung et al., 2020; Reichstein et al., 2019; Xu et al., 2018). Compared to physical models, ML technology can fluently and accurately establish non-linear and complex relationships between diverse independent variables (Koppa et al., 2022; Nearing et al., 2018). Thus, ML-based approaches can create beneficial pathways for knowledge discovery in process models based on extensible data (Moosavi et al., 2021; Reichstein et al., 2019). The main improvements are focused on model approximation, parameterization, bias correction, and hybrid modeling (Brajard et al., 2020; He et al., 2022; Jia et al., 2021; Xu et al., 2014; Zhao et al., 2019). Several studies





have shown that the integration of the DA and ML methods can enhance the reliability of predictions and reduce simulation errors by including physical information in observed data (Brajard et al., 2020; Buizza et al., 2022; Forman and Xue, 2017; Gottwald and Reich, 2021; He et al., 2022). Forman and Xue (2017) integrated a ML model (as a measurement operator) into a DA system to improve the estimation of snow water equivalent. Zhao et al. (2019) used a physics-constrained ML method to improve the latent heat fluxes estimates. He et al. (2022) proposed a hybrid model that can integrate remotely sensed LAI and multi-source SM observations to improve the estimation of ET within the coupled DA and ML framework.

The Heihe River Basin (HRB) is a typical endorheic river basin in the arid and semi-arid regions of Northwest China (Li et al., 2013). The upstream mountain region is mainly covered by alpine meadows, has a complex topography, and receives abundant precipitation. The midstream oasis of the HRB is mainly characterized by irrigated croplands, while the downstream oasis is characterized by riparian forests and tamarisks, and at the periphery of the oasis is vast desert (Xu et al., 2020). In the HRB, precipitation is the main water resource input in mountainous areas and determines the growth of vegetation in the oasis region, as well as supporting urban and population development (Li et al., 2018, 2021). Precipitation in the upstream mountains and irrigation in the midstream oasis can affect mountain runoff, SM, evaporation, and groundwater table in the mid- and downstream oases. Strong land-atmosphere interactions in the HRB affect the water and energy exchange between the surface and atmosphere and influence the sustainability of the oasis (Gao et al., 2008; Pan et al., 2021b). The oasis-desert local circulation in the HRB can lead to the local microclimate features in oasis-desert areas, which include the cooling effect and wind shield effect of the oasis, and the humidity inversion effect within the surrounding desert (Liu et al., 2020).

In recent decades, several comprehensive experiments have been implemented over the HRB to study land-atmosphere interactions, including the Heihe River Basin Field Experiment (Hu et al., 1994), Watershed Allied Telemetry Experimental Research (WATER) (Li et al., 2009), and Heihe Watershed Allied Telemetry Experimental Research (HiWATER) (Li et al., 2013). In recent years, many mesoscale climate models and high-resolution computational fluid dynamics (CFD) models have been used to analyze the effects of land surface conditions on the regional climate (Liu et al., 2018a, 2020; Xie et al., 2018; Zhang et al., 2017a, 2021a). Zhang et al. (2017a) added an irrigation scheme to the WRF model and identified strong cooling and wetting effects on irrigated cropland in the midstream of the HRB. Liu et al. (2020) investigated the oasis-desert microclimate effects based on an improved CFD model and found that the oasis had a cold and wet island effect and wind shield effect. Zhang et al. (2021b) applied the WRF-Hydro model in the HRB and emphasized the role of lateral flow in the regional precipitation circulation. These studies illustrate that mesoscale climate models can be used as essential tools to better understand regional climate and land-atmosphere interactions in the HRB. However, the advantages of improving soil and vegetation processes in regional climate via the coupled DA and ML framework have not been fully exploited.

The goals of this study were to (1) couple the hybrid DA and ML (DA-ML) framework to the WRF model and improve the estimation of LAI and SM; (2) validate the air temperature and specific humidity estimates of the WRF (DA-ML) in the HRB and compare the ET estimates with the ML-based watershed ET; (3) evaluate the effects of the hybrid framework on


near-surface air conditions and land-atmosphere interactions in the mid- and downstream oasis regions; and (4) discuss the effects of the hybrid framework on water vapor flux transport and precipitation in the HRB.

## 2 Study area and dataset

The HRB (37.7°–42.7 °N, 97.1°–102.0 °E) is the second largest endorheic river basin in Northwest China and has an area of approximately 143,000 km², and the elevation ranges from 900 to 5000 m (Figure 1). The annual precipitation is approximately 400 mm, and it gradually decreases from the upstream region of the HRB (south) to the downstream region (north). Land cover types exhibit spatial zonation in the HRB. The upstream region is a typical mountainous environment, including extensive alpine meadows, a few Qinghai spruces, glaciers, and snow. The midstream region is spatially composed of oasis-desert ecosystems, and irrigated cropland is the main component of the oasis in this area. The downstream region of the HRB is covered mainly by desert and riparian ecosystems (riparian forests and tamarisks). Water vapor transport in this region is predominantly controlled by mid-latitude westerly and polar northerly winds (Pan et al., 2021b). As a result, precipitation over the HRB shows strong spatial variability, with more than 70% occurring in the upstream mountains (Wang et al., 2018a, b). Nine meteorological stations were selected for comparison with the model results (Figure 1). Among them, the Arou, Dashalong, and Hulugou stations in the upstream region of the HRB are covered by alpine meadows while the Daman, Wetland, and Huaizhaizi stations in the midstream region are covered by irrigated cropland, wetlands, and barren land, respectively. The Sidaoqiao, Mixed forest, and Desert stations in the downstream regions are covered by tamarisk, riparian forest, and desert, respectively. More details on the *in situ* information and measurement instruments can be found in Chen et al. (2014), Li et al. (2013), and Liu et al. (2018b).

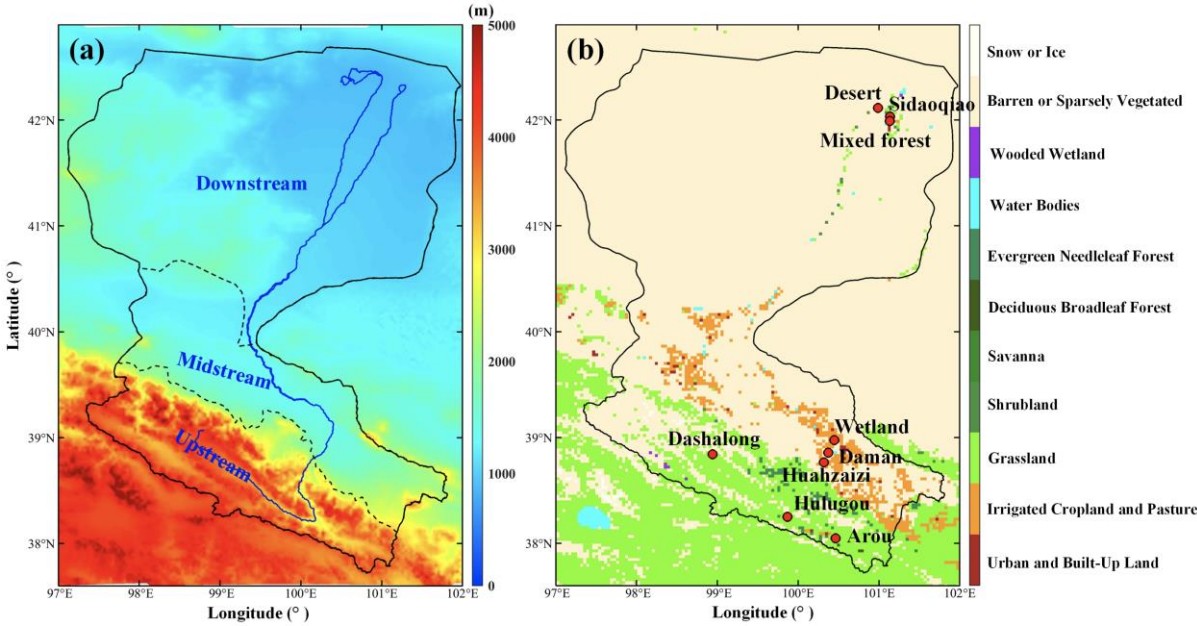



**Figure 1: (a) Land surface elevation, (b) meteorological stations and land cover types in the study area. The solid black line represents the boundary of the HRB.**

To provide a high-resolution land cover and soil texture dataset that matched the WRF simulation period, the regional land cover and soil texture product generated by Zhong et al. (2014) and Song et al. (2016) with a spatial resolution of 30 m was employed. These datasets were downloaded from the National Tibetan Plateau Data Center (TPDC) (Pan et al., 2021a) (https://data.tpdc.ac.cn/en/). The elevation data were generated by NASA's Shuttle Radar Topography Mission (SRTM3, 90 m), which was obtained from the Geospatial Data Cloud (http://www.gscloud.cn/). The assimilated LAI data were retrieved from the Global Land Surface Satellite (GLASS) product with a spatial resolution of 1 km (Xiao et al., 2014; http://www.glass.umd.edu/). Daily LAI observations were generated by linearly interpolating the original 8-day GLASS LAI product. The daily Soil Moisture Active Passive (SMAP) SM product (https://appeears.earthdatacloud.nasa.gov/), with a spatial resolution of 9 km, was integrated into the hybrid framework. In this study, SM observations from the eco-hydrological wireless sensor networks (WATERNET) in the up- and midstream of the HRB are used to validate the SM estimates from the WRF (DA-ML). The validation SM dataset in the upstream regions was mainly covered by grassland and obtained by averaging SM observations from 40 nodes. There are 9 network nodes in the midstream cropland that measured SM at the depths of 10 cm every 5 minutes (Che et al., 2019; Jin et al., 2014) (https://data.tpdc.ac.cn/en/). ETMap is a ML-based watershed ET product (daily/1 km) based on eddy covariance (EC) observations, remote sensing data, micrometeorological data, and the random forest method (Xu et al., 2018). In this study, nine automatic weather station (AWS) observations were used to validate the WRF simulations (https://data.tpdc.ac.cn/en/). The data recorded at these stations were obtained from the "Heihe Watershed Allied Telemetry Experimental Research" (HiWATER) experiment released by the TPDC (Liu et al., 2018b). The locations of the stations are shown in Figure 1. Among them, the Hulugou weather station observations are produced by Chen et al. (2014). Precipitation datasets from the China Meteorological Forcing Dataset (CMFD) (He et al., 2020a) and atmospheric forcing data (AFD) in the HRB (Pan et al., 2014) were compared with the WRF (DA-ML) simulations. These datasets were obtained from the TPDC. The CMFD generates a gridded meteorological dataset with a spatial resolution of 0.1° and a temporal resolution of 3 h by fusing observations from 740 operational stations of the China Meteorological Administration. Pan et al. (2014) generated gridded atmospheric forcing data using the WRF model over the HRB at an hourly 0.05° resolution. These datasets have been widely used as input data for various models as well as for environmental and climate change analyses (Xu et al., 2019; Zhang et al., 2016).

## 3 Methodology

### 3.1 WRF model setup

The advanced research WRF model version 4.0.3 (Skamarock et al., 2019) was used in this study. The WRF is a state-of-the-art numerical weather and climate model designed by the National Center for Atmospheric Research (NCAR) for meteorological research and numerical weather predictions (Wang et al., 2021). The model source code is available at the





official repository for WRF (https://github.com/wrf-model/WRF). The model domain covering the HRB consisted of two-
way nested domains with 9 km and 3 km grid spacing. Only the simulation results for the 3 km grid were used in this study
(Figure 1). This high-resolution setting excludes the uncertainty of the cumulus parameterization and thus simulates the soil-
precipitation feedback more realistically (Prein et al., 2015). In the vertical direction, 28 vertical sigma levels from the
surface to 50 hPa were used. The atmospheric lateral boundary conditions in the WRF model were provided by the ERA5
reanalysis data set with a 0.25° spatial resolution and hourly temporal resolution
(https://cds.climate.copernicus.eu/cdsapp#!/search?type=dataset). It is widely used in WRF simulations and provides
boundary and initial conditions (Liu et al., 2021; Ma et al., 2022). The physical parameterization schemes selected in this
study included the Rapid Radiative Transfer Model longwave and shortwave radiation scheme (Mlawer et al. 1997),
Thompson microphysics scheme (Thompson et al., 2008), Mellor–Yamada–Janjic planetary boundary layer scheme (Janjic,
1994), and Noah-MP land surface scheme (Yang et al., 2011). The time step was 30 s, and the time resolution of the model
output was hourly. Further details regarding the WRF model setup are presented in Table 1. The dynamic vegetation
parameterization scheme was turned on in the Noah-MP model to generate dynamic LAI simulations. We resampled the
spatial resolution of the WRF simulations to 1 km with the bilinear interpolation method for comparison with the station
observations.

**Table 1: WRF model setup.**

| Simulation period | May to September, 2015 |
|---|---|
| Horizontal grid spacing | 9 km (100 × 181 grid points), 3 km (181 × 220 grid points) |
| Vertical levels | 28 sigma-levels |
| Forcing data | ERA5 (0.25°, hourly) |
| Spin-up time | April (one month) |
| Longwave radiation | RRTM scheme (Mlawer et al., 1997) |
| Shortwave radiation | RRTM scheme (Mlawer et al., 1997) |
| Microphysics (MP) | Thompson scheme (Thompson et al., 2008) |
| Planetary boundary layer (PBL) | Mellor–Yamada–Janjic scheme (Janjic, 1994) |
| Surface layer | Eta similarity scheme (Janjic, 1994) |
| Land surface model (LSM) | Noah-MP land surface model (Yang et al., 2011) |

**3.2 Hybrid model**

In this study, the hybrid model proposed by He et al. (2022) based on the DA and ML methods was incorporated into
the WRF model to improve the LAI, SM, and ET simulations. The hybrid approach relies on the DA method to update the
vegetation dynamics of the Noah-MP model and the ML method to construct a three-layer SM surrogate model.

In the DA part, the remotely sensed LAI was assimilated using the ensemble Kalman filter (EnKF) method to update
the leaf biomass (LFMASS) and optimize the specific leaf area (SLA) in the Noah-MP model. LAI is estimated as the





product of leaf biomass predictions and SLA (LAI = LFMASS × SLA) in the Noah-MP model. Model ensembles were generated by adding normally distributed random errors to the model states (LFMASS) and parameters (SLA). The ensemble size is set as 40 to ensure an accurate approximation of the error covariances while maintaining computational efficiency (Seo et al., 2021). Normally distributed errors with a mean of zero and a standard deviation of 10 g m$^{-2}$ were added to the

LFMASS (Ahmad et al., 2022; Xu et al., 2021). The standard deviation of the SLA was set to 10% of the default parameter (Xu et al., 2021). Further, a uniform observation error standard deviation of 0.1 (-) was added to the remotely sensed LAI (He et al., 2022; Xu et al., 2021). These relevant statistical values have been widely used in previous LAI DA studies (Ahmad et al., 2022; Ling et al., 2019; Rahman et al., 2022).

In the ML part, the standardized soil texture, land cover, air temperature and humidity, wind speed, rainfall, solar

radiation, LAI, and SM observations were used to construct the SM surrogate model. The extreme gradient boosting (XGBoost) method was used to integrate *in situ* SM profile observations (from 19 automatic weather stations) and SMAP SM products to improve multi-layer SM simulations. The first layer (the top 0.1 m) of *in situ* SM observations and SMAP SM were trained to establish the surface layer ML model. The averaged second (0.1–0.4 m) and third (0.4–1.0 m) layers of *in situ* SM observations were used to construct the root zone ML models. The SM observations at different depths were

averaged to be consistent with the Noah-MP model soil layer. The number of SM training samples in the first, second, and third layers are 9824, 7804, and 7793, respectively. A ten-fold testing method is employed to examine the performance of each ML method. In each fold, 90% of the training samples are used to train the model, and the remaining 10% of the data is used to test the model. The results demonstrate that the SM surrogate model can consider the patterns of midstream irrigation and the downstream groundwater table in the HRB and improve the ET estimates of the Noah-MP. More details regarding

this method can be found in He et al. (2022).

The coupled land-atmosphere DA-ML system consists of two steps. In the first step, the meteorological forcing data were generated from the WRF model at time *t*. Then, the meteorological forcing data and initial states were input into the Noah-MP model to simulate the LAI, SM, and ET. In the second step, the hybrid DA-ML method was used to update the LAI and SM estimates in the Noah-MP model at time *t* + 1. The updated soil and vegetation conditions were re-fed into the

WRF model and affected the atmospheric structure and state. Eventually, the WRF model and DA-ML method were coupled and run dynamically and consistently through the cycles of steps one and two.

Two experiments were conducted using the WRF model to investigate the effects of the DA-ML method. These two experiments consisted of an experiment under natural conditions without DA and ML [WRF (OL)], and another experiment implementing DA and ML [WRF (DA-ML)]. The simulation covered the period from April 1 to September 30, 2015, and

the first month was used for the spin-up. The computation details about the WRF (OL) and WRF (DA-ML) are shown in Table A1. The differences between the WRF (DA-ML) and WRF (OL) simulations were used to investigate the effects of LAI and SM integration. The root mean square deviation (RMSD) and coefficient of determination (R$^2$) statistical metrics were used to evaluate the performance of the WRF (DA-ML).





## 4 Results and discussion

**4.1 LAI, soil moisture, and evapotranspiration**

Figure 2 shows the monthly averaged LAI estimates from the WRF (OL), WRF (DA-ML), and GLASS products. As indicated, the WRF model failed to capture the magnitude and seasonality of the LAI. This is because the simulation of LAI dynamics in Noah-MP is controlled by the planting date, harvest date, and growing degree days in the cropland (Liu et al., 2016). In addition, inaccurate specification of the SM saturation, Vcmax, and Clapp-Hornberger $b$ parameter affects
photosynthesis and biomass accumulation in vegetation (Cuntz et al., 2016; Levis et al., 2012). All these parameters are site-specific and empirical, and cannot be easily applied across regions. The assimilation systematically increased LAI during the growing season, and a significant increment in LAI was observed in mid-summer (June-August). The seasonal pattern of the WRF (DA-ML) was more consistent with that of the GLASS LAI than the WRF model, which indicates that the WRF (DA-ML) provides essential information for modeling vegetation dynamics. The simulated LAIs in the cropland, grassland, forest,
and shrubland areas were 1.12, 1.05, 1.49, and 0.33 $m^2$ $m^{-2}$, respectively, all of which were lower than that of the GLASS LAI. After assimilation, the simulated bias of the LAI in the HRB can be reduced from 0.94 to 0.11 $m^2$ $m^{-2}$.

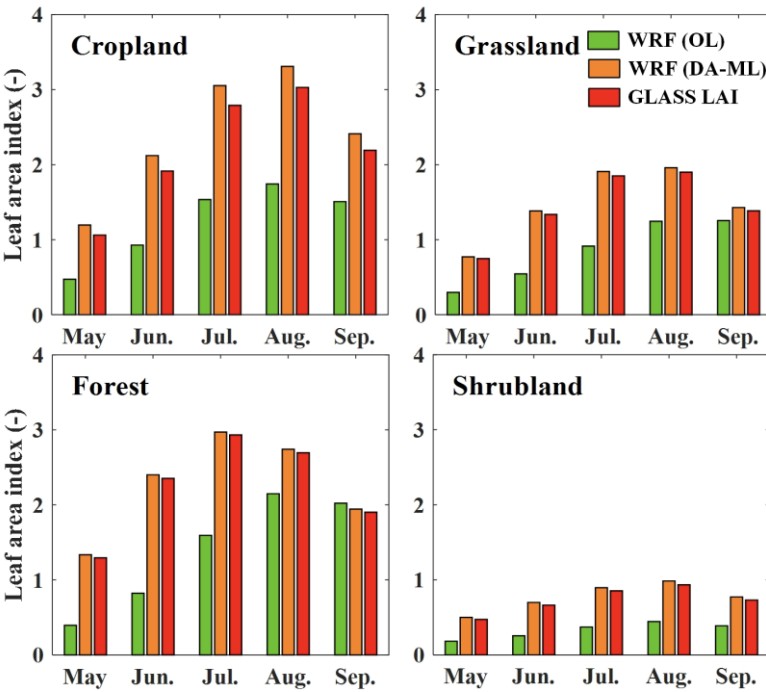

**Figure 2: Seasonal variations of the LAI estimates for cropland, grassland, forest, and shrubland in the HRB.**

The SM estimates from the WRF (OL) and WRF (DA-ML) are validated over the up- and midstream WATERNET in
Figure 3. The SM estimates from the WRF model were markedly lower than WATERNET observations for cropland because the impacts of irrigation events on SM estimates are not fully considered in the Noah-MP model (He et al., 2022;





Zhang et al., 2020). The Noah-MP model also slightly underestimates SM in the upstream regions because it ignores the effects of dense root systems and soil organic matter on SM estimation (Chen et al., 2012; Sun et al., 2021). As anticipated, SM predictions from the WRF (DA-ML) are closer to the measurements than those of WRF. WRF (DA-ML) SM retrievals

indicate a reasonable response to the precipitation and irrigation events in the midstream cropland. Similarly, the WRF (DA-ML) SM dynamics show a characteristic response to precipitation in the upstream regions. In general, the WRF (DA-ML) can use the information contained in remotely sensed LAI and multi-source SM observations to improve land surface conditions.

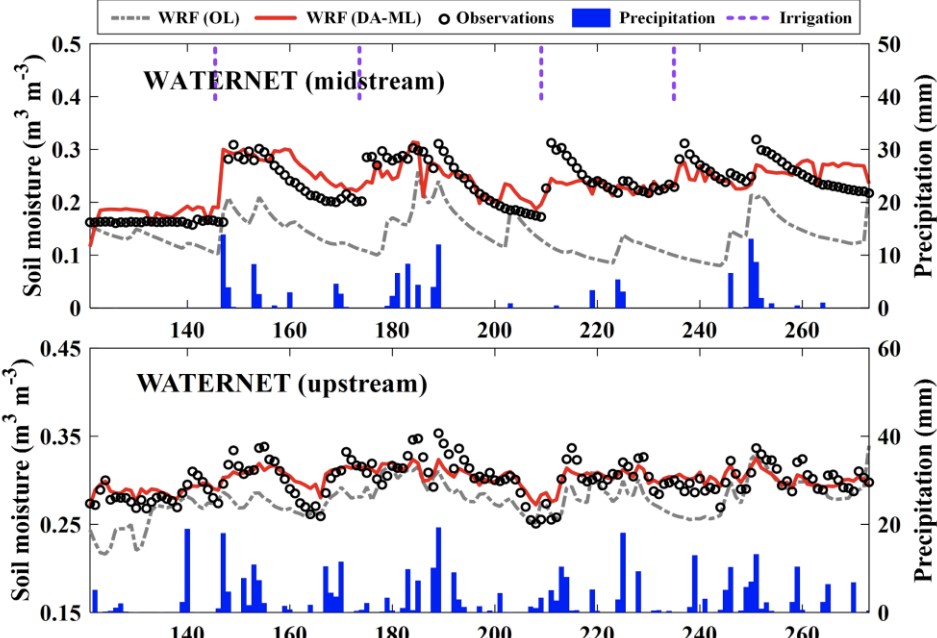

**Figure 3: The time series of SM estimates from the WRF (OL) and WRF (DA-ML) models against WATERNET observations in 2015.**

Figure 4 shows the spatial patterns of the averaged LAI and SM estimates from May to September 2015. The WRF simulation significantly underestimated the LAI, particularly in the up- and midstream vegetation areas of the HRB. In addition, it underestimated the SM in the mid- and downstream vegetation regions. The integration of LAI and SM into the

WRF model improved the estimation of LFMASS and SM, and increased LAI and SM in the HRB. The maps of estimated LAI and SM from the DA-ML method consistently resembled the rainfall, vegetation cover, irrigation event, and shallow groundwater table features (Xu et al., 2018, 2020). Figure 4 also shows the LAI and SM differences between the WRF (DA-ML) and WRF (OL) simulations. The maximum LAI (SM) difference from the WRF (DA-ML) minus the WRF (OL) reaches approximately 2.24 $m^2 m^{-2}$ (0.16 $m^3 m^{-3}$) and is present in the midstream oasis of HRB. The difference between the

WRF (OL) and WRF (DA-ML) was due to the effects of irrigation and crop growth. Similarly, this difference in the downstream oasis is a result of the shallow water table as well as the growth of tamarisk and riparian forests. In the upstream alpine meadows, the LAI simulated by the WRF (DA-ML) was greatly increased compared to that of the WRF (OL).





However, the SM enhancement in the WRF (DA-ML) was not significant due to the sufficient precipitation in mountainous areas.

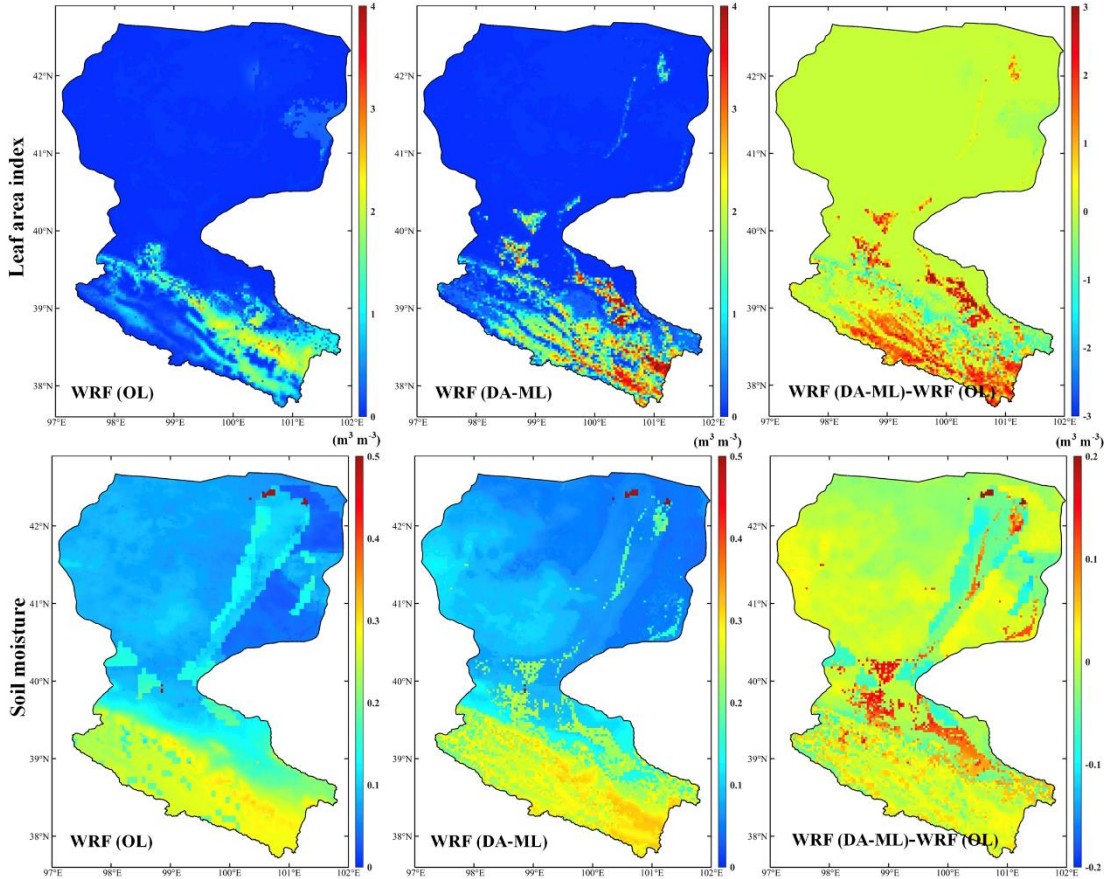

**Figure 4: The LAI and SM estimates from the WRF (OL) and WRF (DA-ML) models during the growing season in 2015, and the average difference in the LAI and SM between the WRF (DA-ML) and WRF (OL) [WRF (DA-ML) minus WRF (OL)].**

Figure 5 shows the spatial distribution of ET estimates from the WRF (OL), WRF (DA-ML), and ETMap over the HRB. The results indicate that the ET values from the WRF model were underestimated, especially in the midstream oasis region, which was mainly because the WRF model underestimated the SM and LAI (see Figure 2 and 3) during the growing season. Compared with the WRF (OL) model, the WRF (DA-ML) method improves the estimation of ET and the spatial distribution is consistent with that of ETMap because of the effective information contained in the remote sensing LAI and multi-source SM observations. The spatial patterns of ET from the DA-ML method showed a significant gradient from wet to dry owing to variations in precipitation and vegetation cover. In the upstream regions of the HRB, the spatial pattern of retrieved ET was mainly controlled by precipitation and vegetation cover. The ET values were higher in areas with heavier precipitation and denser vegetation. In the midstream region, the spatial pattern of ET was well aligned with the oasis caused by crop growth and irrigation. Meanwhile, the ET values were higher in the downstream oasis because of shallow water tables and

transpiration from riparian forests (Xu et al., 2018, 2020). The sparsely vegetated areas covered by desert and Gobi in the mid- and downstream regions had the lowest ET values. The results show that the integration of remotely sensed LAI and 260 multi-source SM observations is essential for studying land-atmosphere water vapor fluxes (ET) because of the realistic land surface conditions.

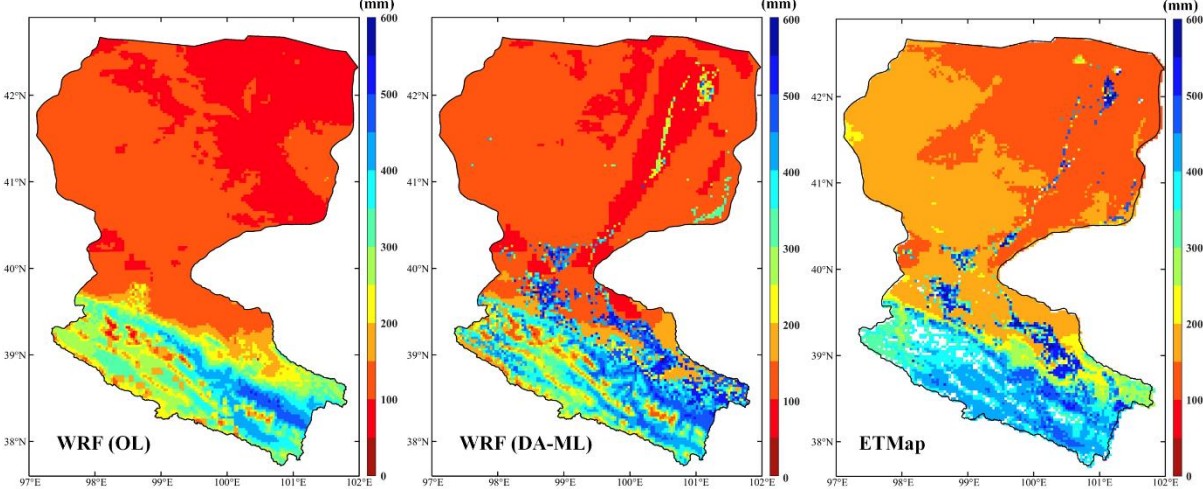

**Figure 5: Spatial distribution of evapotranspiration estimates obtained from the WRF (OL), WRF (DA-ML), and ETMap during the growing season in 2015.**

## 4.2 Air temperature and specific humidity

The monthly averaged seasonal cycle of the estimated air temperature and specific humidity from the WRF (OL), WRF (DA-ML), and observations at nine sites are shown in Figure 6 and 7. As indicated, the WRF model overestimated (underestimated) the air temperature (specific humidity) in the HRB, especially in the midstream oasis (Daman and Wetland stations), which was mainly because the WRF model underestimated the SM and LAI (see Figure 2 and 3) in the HRB.
Compared to the WRF model, the WRF (DA-ML) simulated seasonal cycles of air temperature and specific humidity at the nine sites were closer to the measurements, which was because the integration of remotely sensed LAI and multi-source SM observations improves the estimation of vegetation dynamics and SM, decreases the air temperature, and increases the specific humidity. The increased specific humidity was due to the enhanced evaporation from the soil and stronger transpiration from the expanded vegetation cover. Simultaneously, evaporation absorbs a large amount of energy, thereby 275 reducing the air temperature (Wen et al., 2012). The discrepancy between the WRF (OL) and WRF (DA-ML) was amplified in the middle of the growing season (June, July, and August) due to vegetation growth and irrigation events. After integrating LAI and SM, the simulated air temperature and specific humidity values from the Daman station decreased and increased by approximately 1.75 K and 1.86 g kg$^{-1}$, respectively. But for Sidaoqiao, the air temperature and specific humidity decrease and increase by about 0.59 K and 0.41 g kg$^{-1}$, respectively. The estimated air temperature and specific humidity





increased from May to July and decreased from August to September. The specific humidity estimated at Daman exhibited

significant seasonal variations due to irrigation events, crop planting, and harvesting.

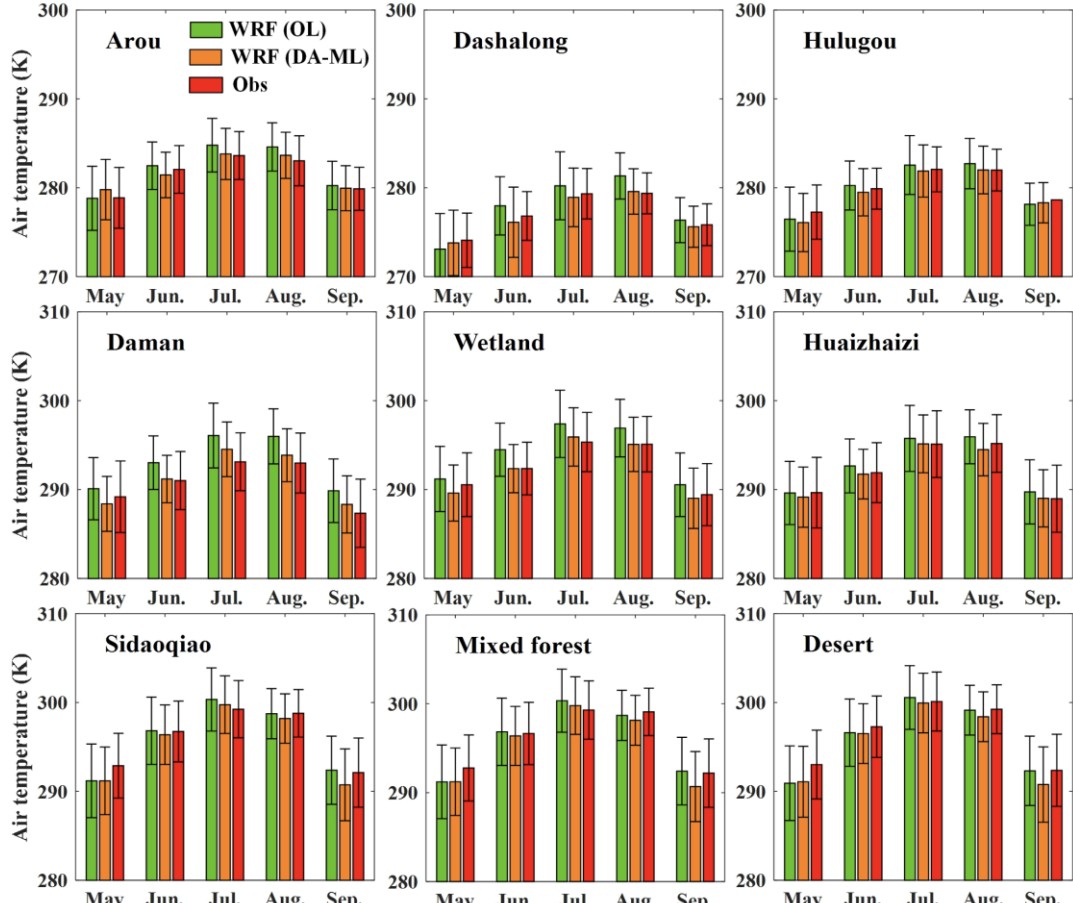

**Figure 6: Monthly averaged air temperature simulations from the WRF (OL) and WRF (DA-ML) versus the observations at nine sites in 2015 (error range denotes the standard deviation).**





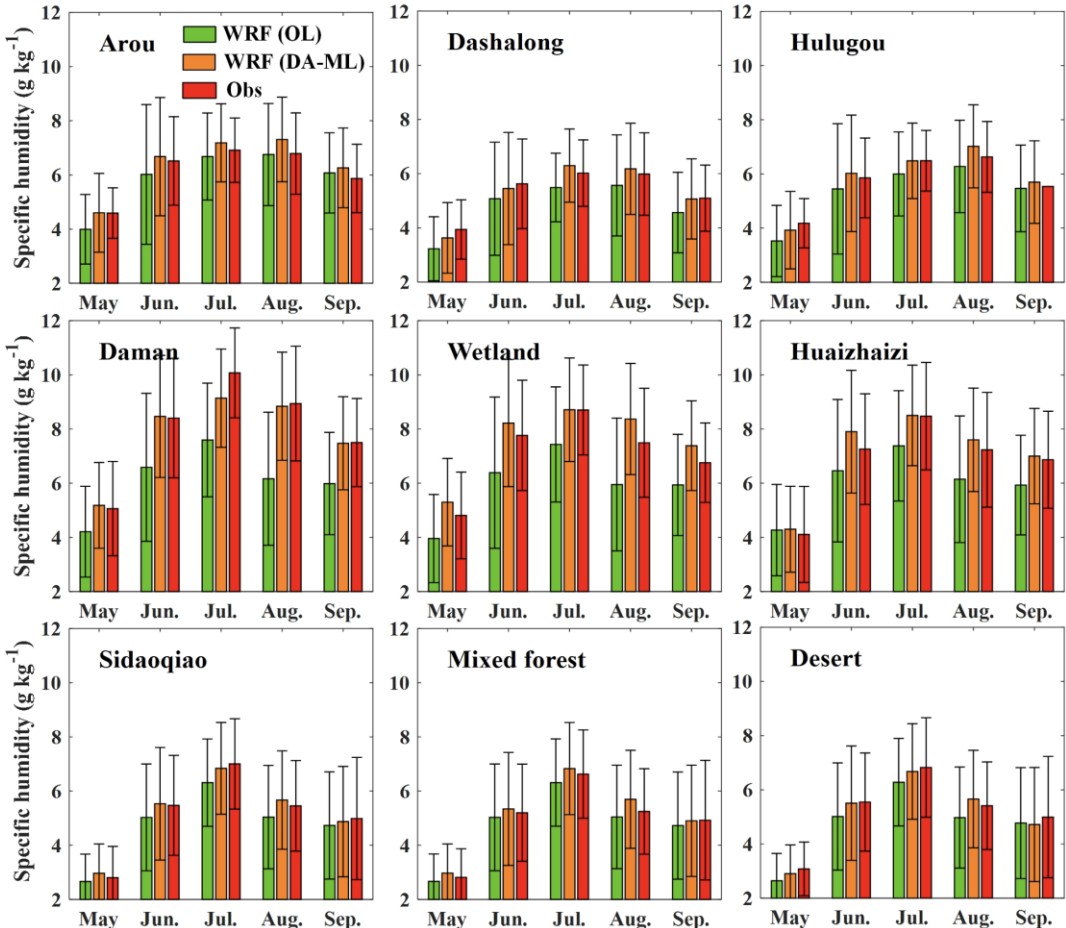

**Figure 7: Monthly averaged specific humidity simulations from the WRF (OL) and WRF (DA-ML) versus the observations at nine sites in 2015 (error range denotes the standard deviation).**

Table 2 and 3 further compare the simulated air temperature and specific humidity with the same variables from the station observations. The WRF (OL) results show a dry bias in the HRB region, which is reduced by the simulation of the WRF (DA-ML). The statistical metrics (i.e., $R^2$ and RMSD) of the daily air temperature and specific humidity estimates from the WRF (OL) and WRF (DA-ML) methods are shown in Table 2 and 3. For the nine sites, the average RMSD of the air temperature (specific humidity) estimates from the WRF (DA-ML) was 1.41 K (0.82 g kg$^{-1}$), which was 21.23% (24.07%) lower than the RMSD of 1.79 K (1.08 g kg$^{-1}$) from the WRF model.

**Table 2: Averaged air temperature and $R^2$ and RMSD of the WRF (OL) and WRF (DA-ML) compared with the measurements at the nine sites.**

| Study site | Obs | WRF (OL) | | | WRF (DA-ML) | | |
|---|---|---|---|---|---|---|---|
| | | Sim | $R^2$ (-) | RMSD (K) | Sim | $R^2$ (-) | RMSD (K) |
| Arou | 281.48 | 282.15 | 0.87 | 1.48 | 281.44 | 0.89 | 1.11 |





| | | | | | | | |
|---|---|---|---|---|---|---|---|
| Dashalong | 276.32 | 277.76 | 0.84 | 1.97 | 276.56 | 0.86 | 1.75 |
| Hulugou | 279.01 | 280.06 | 0.86 | 1.33 | 279.05 | 0.90 | 1.21 |
| Daman | 290.75 | 292.91 | 0.84 | 2.36 | 291.16 | 0.88 | 1.52 |
| Wetland | 292.49 | 293.99 | 0.90 | 1.98 | 292.29 | 0.92 | 1.31 |
| Huazhaizi | 292.09 | 292.64 | 0.91 | 1.33 | 291.94 | 0.94 | 1.12 |
| Sidaoqiao | 295.84 | 296.12 | 0.92 | 1.80 | 295.53 | 0.94 | 1.47 |
| Mixed forest | 295.68 | 296.31 | 0.92 | 1.83 | 295.32 | 0.93 | 1.56 |
| Desert | 296.30 | 296.79 | 0.93 | 2.04 | 295.77 | 0.95 | 1.61 |
| Average | 288.88 | 289.85 | 0.89 | 1.79 | 288.78 | 0.91 | 1.41 |

**Table 3: Averaged specific humidity and $R^2$ and RMSD of the WRF (OL) and WRF (DA-ML) compared with the measurements at the nine sites.**

| Study site | | WRF (OL) | | | WRF (DA-ML) | | |
|---|---|---|---|---|---|---|---|
| | Obs | Sim | $R^2$ (-) | RMSD (g kg$^{-1}$) | Sim | $R^2$ (-) | RMSD (g kg$^{-1}$) |
| Arou | 6.13 | 5.90 | 0.85 | 0.88 | 6.31 | 0.88 | 0.78 |
| Dashalong | 5.33 | 4.78 | 0.87 | 0.83 | 5.42 | 0.88 | 0.71 |
| Hulugou | 5.74 | 5.34 | 0.82 | 0.82 | 5.81 | 0.85 | 0.78 |
| Daman | 7.97 | 6.12 | 0.74 | 2.26 | 7.98 | 0.82 | 1.04 |
| Wetland | 7.09 | 5.94 | 0.81 | 1.55 | 7.57 | 0.84 | 1.03 |
| Huazhaizi | 6.78 | 6.05 | 0.79 | 1.25 | 7.13 | 0.84 | 1.10 |
| Sidaoqiao | 5.14 | 4.76 | 0.91 | 0.73 | 5.17 | 0.91 | 0.64 |
| Mixed forest | 4.96 | 4.75 | 0.90 | 0.68 | 5.09 | 0.91 | 0.63 |
| Desert | 5.17 | 4.73 | 0.91 | 0.75 | 5.09 | 0.91 | 0.65 |
| Average | 6.03 | 5.37 | 0.84 | 1.08 | 6.17 | 0.87 | 0.82 |

Figure 8 compares the spatial patterns of the air temperature and specific humidity maps from the WRF (OL) and WRF (DA-ML). Compared with the WRF (OL), significant differences were observed in the WRF (DA-ML). The integration of LAI and SM decreases air temperature and increases specific humidity in the vegetated area of the HRB, particularly in the midstream oasis region. The spatial distribution of specific humidity from the WRF (DA-ML) is consistent with the LAI and SM maps in Figure 4 and the ET map in Figure 5. The results show that the improved LAI and SM simulations make the oasis a typical wet and cold island compared with the surrounding desert. Higher air humidity was generated above the vegetated areas, whereas lower specific humidity occurred in desert areas. The significant wetting and cooling effects propagate in desert areas to a maximum distance of approximately 10-20 km from the edge of the oasis. In the midstream oasis, the dominant vegetation is irrigated cropland, and the vegetation cover was only approximately 42% in the original WRF model; however, the vegetation cover was updated to approximately 70% in the WRF (DA-ML). The results indicate that the existence of the oasis can produce wetting and cooling effects in the surrounding desert because oasis-desert interactions create a water vapor flux from the oasis to the surrounding desert. This transport process is beneficial for





increasing desert water vapor and maintaining the sustainability of desert ecosystems (Li et al., 2016; Liu et al., 2020). A similar pattern was observed in the downstream oasis. However, because of the different vegetation types and decreased SM and vegetation cover (Figure 4), the downstream oasis exhibited a weaker wet island effect. The results also indicated that enhanced vegetation transpiration increases specific humidity and reduces air temperature owing to increased LAI in the

upstream region of the HRB.

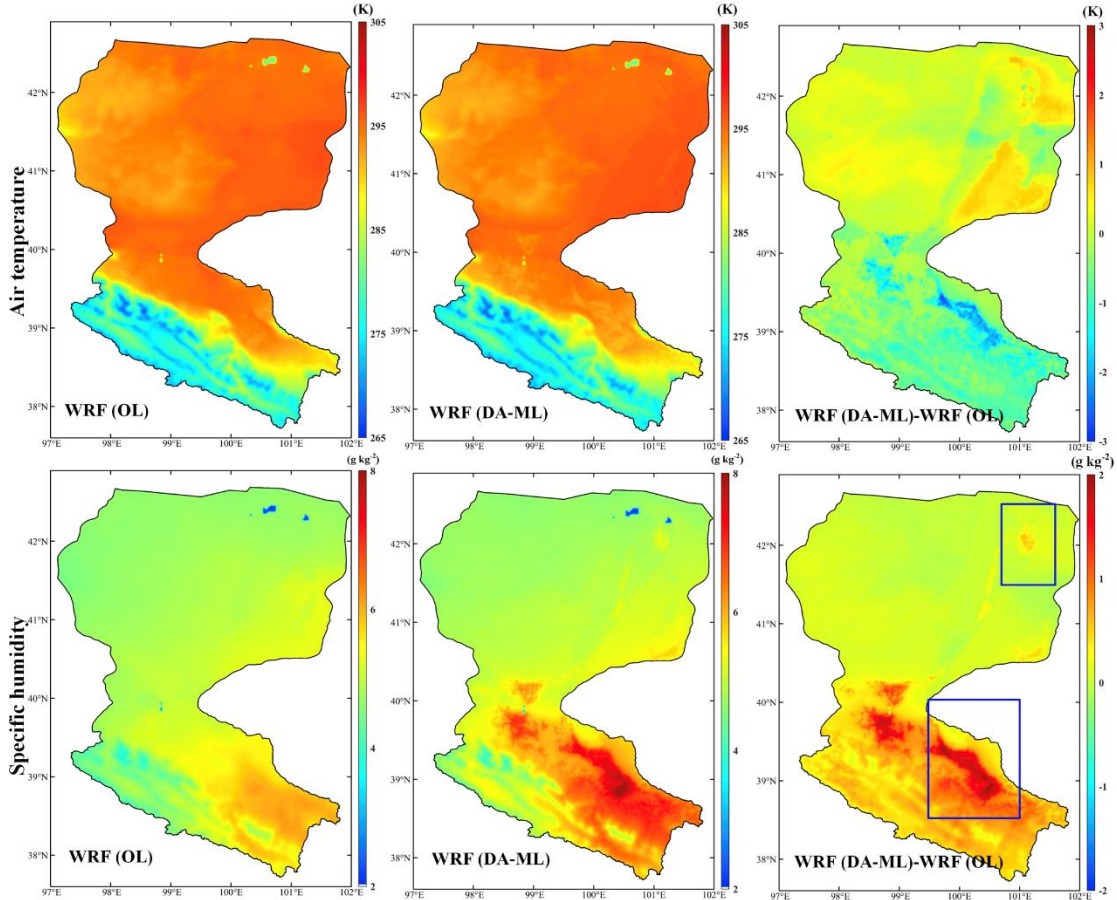

**Figure 8: Spatial distribution of air temperature and specific humidity estimates from the WRF (OL) and WRF (DA-ML) during the growing season in 2015, and the average difference in air temperature and specific humidity between the WRF (DA-ML) and WRF (OL) [WRF (DA-ML) minus WRF (OL)].**

The wetting and cooling effects of the oasis are mainly present at the bottom of the atmosphere, and their magnitude is closely and positively related to the difference between LAI and SM. Two rectangular areas (blue rectangle) were selected in Figure 8 to further analyze the effect of the DA-ML on the local climate in the mid- and downstream areas. As illustrated in Figure 9, the wetting and cooling effects of the midstream oasis were the strongest in the southern irrigated cropland and gradually decreased in the northern desert areas. The magnitudes of the surface wetting and cooling effects were consistent

with the differences in LAI and SM estimates from the WRF (DA-ML) and WRF (OL). For example, the difference in SM



peaks at 38.8°N and the wetting and cooling effects of midstream irrigated cropland were also stronger in this region. These results suggest that the wetting and cooling effects caused by irrigation and vegetation growth occur mainly in the oasis region and do not affect more distant non-oasis areas. Similarly, the higher specific humidity and lower air temperature in mountainous areas may be due to the wetting and cooling effects of evaporation from grassland. Thus, these effects in the
oasis region are also affected by mountain winds owing to the increased altitude and decreased air temperature (Zhang et al., 2017b). Moreover, the wetting and cooling effects of the oasis were mainly concentrated in the boundary layer, gradually decreased from the land surface upward, and disappeared at the height of approximately 600 hPa (4000 m). Similar results have been demonstrated in several previous studies (Liu et al., 2020; Wen et al., 2012; Zhang et al., 2017a, b).

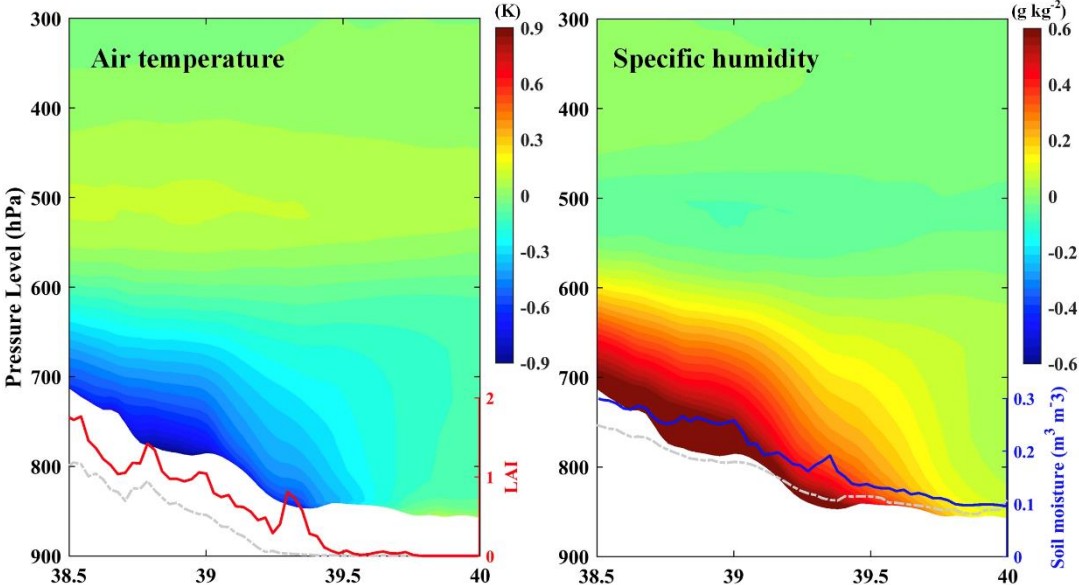

**Figure 9: Mean vertical profile of differences in air temperature and specific humidity between the WRF (DA-ML) and WRF (OL) [WRF (DA-ML) minus WRF (OL)] and mean LAI and SM during the growing season in 2015 in the midstream oasis. The dashed and solid lines represent the WRF (OL) and WRF (DA-ML), respectively. The white shaded area represents the change in elevation.**

Figure 10 shows the same wetting and cooling effects in the downstream oasis. Compared to the midstream irrigated
cropland, the downstream oasis wetting and cooling effects were mainly influenced by the growth of riparian forests, tamarisk, and shallow groundwater tables. The wetting and cooling effects showed maximum values at 42°N owing to the strong LAI and SM shifts. The results also indicate that the wetting and cooling effects of the downstream oasis were weaker than those of the midstream oasis. By integrating LAI and SM, the mid- and downstream oasis air temperature decreases by 0.96 and 0.12 K and the specific humidity increases by 0.52 and 0.06 g kg$^{-1}$, respectively. Meanwhile, at the height of
approximately 500 hPa, slight warming and drying effects were observed, which may be related to the movement of air masses owing to meridional circulation (Figure 12). The airflow from the desert brings dry air, which increases the





temperature and decreases the specific humidity. In general, the integration of the LAI and SM data can produce more realistic land surface conditions in the oasis region and lead to stronger wetting and cooling effects.

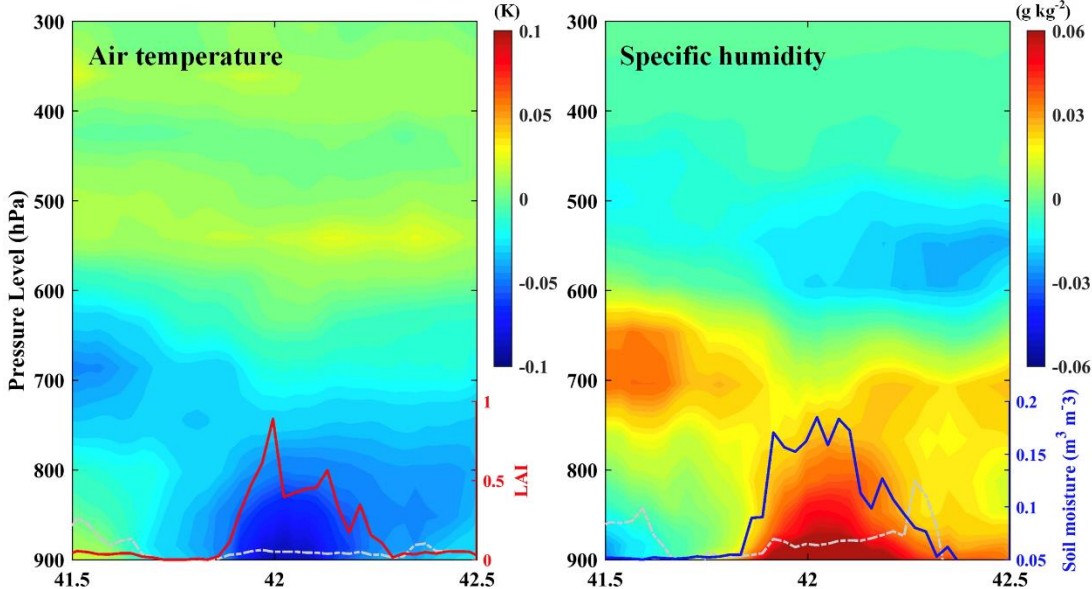

**Figure 10: Same as Figure 9 but for the downstream oasis.**

**4.3 Wind speed and precipitation**

The mean wind vectors at 10 m during the growing season from the WRF (OL) and WRF (DA-ML) in the mid- and downstream oases are shown in Figure 11. By comparing the wind speeds simulated by the WRF (OL) and WRF (DA-ML), we found that irrigation and vegetation growth in the midstream oasis produced a wind-shield effect. The average wind

speed in the midstream oasis was reduced from 1.92 m/s to 1.23 m/s by integrating the LAI and SM. In addition, the wind speed within the oasis was less than that of the surrounding desert because of the drag force of crops, shelterbelts, and residential areas in the oasis changes the wind direction (Liu et al., 2020). In Figure 11, the bulk transfer coefficient ($C_h$) from the WRF (DA-ML) was used to demonstrate the enhanced surface roughness in the oasis. $C_h$ is an important parameter for calculating the heat transfer between the land and atmosphere, and it is mainly related to the length of the surface

roughness and the intensity of the stability of the atmospheric surface layer (Smedman et al., 2007). The results show that the $C_h$ estimates are higher and reduce the wind speed in the midstream oasis compared to the surrounding desert. Ozdogan et al. (2006) and Liu et al. (2020) also showed that irrigation and crop growth enhanced the surface roughness and slow wind speed. In addition, the cooling effects due to irrigation and crop growth in the midstream oasis may lead to atmospheric sinking and drive cold and moist airflow from the oasis to the surrounding desert. As shown in Figure 11, area to the south of

the midstream irrigated cropland is the Qilian Mountains whereas the area to the north is a large desert. Therefore, the southerly airflow generated by the midstream oasis weakened the background north wind. The results also showed that the





mountain-valley winds generated from the Qilian Mountains were enhanced with the improved LAI and SM estimates. This divergent wind direction is opposite to the direction of the northward oasis air flow and the background northerly wind, which reduces the surface wind speed in the desert (approximately 39.0°N). The mean wind vectors in the downstream oasis

are shown in Figure 11. The wind speed in the downstream desert regions is slightly higher than in the oasis regions. The lower wind speed in the oasis is helpful to plant growth, people's survival environment, and the maintenance of the oasis and desert ecosystem (Wang and Cheng, 1999). The results also indicated that the wind vector in the downstream oasis was mainly controlled by the background northerly wind and the effects of LAI and SM integration on the wind vectors were weaker.

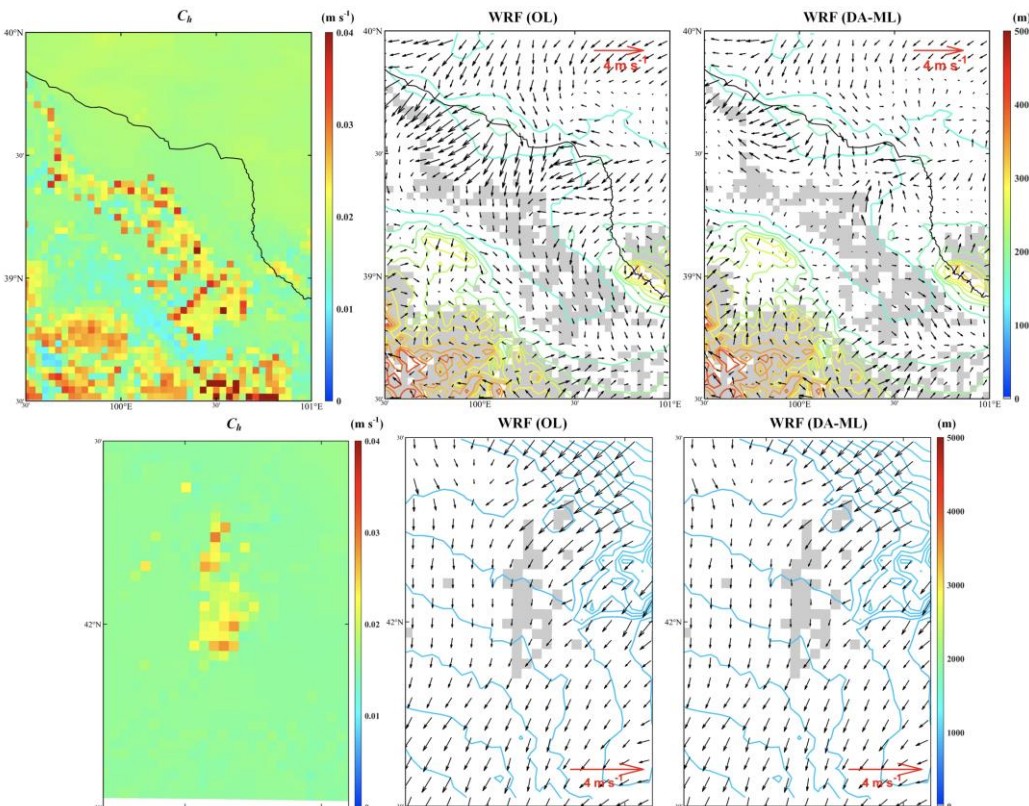


**Figure 11: Mean bulk transfer coefficient ($C_h$) from the WRF (DA-ML) and wind vectors at 10 m during the growing season from the WRF (OL) and WRF (DA-ML) in the midstream (top) and downstream (bottom) oasis. Colored contours indicate elevations above ground level and shading indicates the extent of the oasis.**

         The integration of the LAI and SM affected the wind speed at the land surface and the local circulation through land-

atmosphere interactions. Figure 12 shows the differences in the local meridional circulation in the mid- and downstream oases between the WRF (DA-ML) and WRF (OL). Compared with the flat topography of the downstream oasis, the topography of the midstream oasis generally varies from plains to mountains (from low to high altitude) from north to south. The surface thermal characteristics of the oasis and surrounding desert differed significantly; therefore, strong horizontal





temperature and humidity field gradients were observed at the intersection of the boundary layer of the mountains, oasis, and
surrounding desert (Meng et al., 2015; Wen et al., 2012). As shown in Figure 12, the air humidity and vegetation cover in the
midstream oasis were enhanced by integrating the LAI and SM, which resulted in stronger evaporation from irrigated
cropland than from the surrounding desert. The divergence of the lower atmosphere over the midstream oasis is enhanced,
and the wet and cold air masses are transferred to the surrounding desert through advection, whereas the dry and hot air is
transferred into the oasis from the upper atmosphere (approximately 550 hPa). In the upper atmosphere, air masses enhanced
the background northerly winds (orange areas) and brought more hot and dry air over the oasis, which increased the air
temperatures and decreased the specific humidity over the midstream oasis (Figure 9). The transfer of water vapor from the
oasis to the desert helps to enhance the air humidity of the surrounding desert and maintain the growth of desert vegetation.
At the same time, energy transfer from the desert to the oasis facilitates the enhancement of photosynthesis and transpiration
of the oasis vegetation. The interaction between oases and deserts, enhanced by the integration of LAI and SM, contributes
to the sustainable development of oasis-desert ecosystems (Chu et al., 2005; Liu et al., 2020; Meng et al., 2012).

In addition, warming and drying in the upper atmosphere may have been caused by the strengthening of the mountain
plain circulation due to enhanced vegetation cover. The integration of the LAI and SM produces cooling and wetting effects
in the Qilian Mountain region. The mountain-valley winds generated by mountain plain circulation can extend to the
surrounding oasis and desert regions and influence the oasis-desert interactions (Zhang et al., 2017b). Similar to the
midstream oasis, the downstream oasis creates an advection from the oasis to the desert at the bottom of the atmospheric
boundary layer while hot and dry air masses in the upper atmosphere are transported to the oasis. The air masses transported
from the desert to the oasis reduced the background northerly winds, resulting in a slight decrease in wind speed (at the
height of approximately 650 hPa). However, oasis-desert interactions are weaker in the downstream region than in the
midstream region under actual weather or climate conditions, which is attributed to the local circulation being weakened by
stronger background northerly winds. Overall, the simulation of soil and vegetation characteristics can be improved by
integrating LAI and SM and enhancing land-atmosphere interactions in mid- and downstream oases.





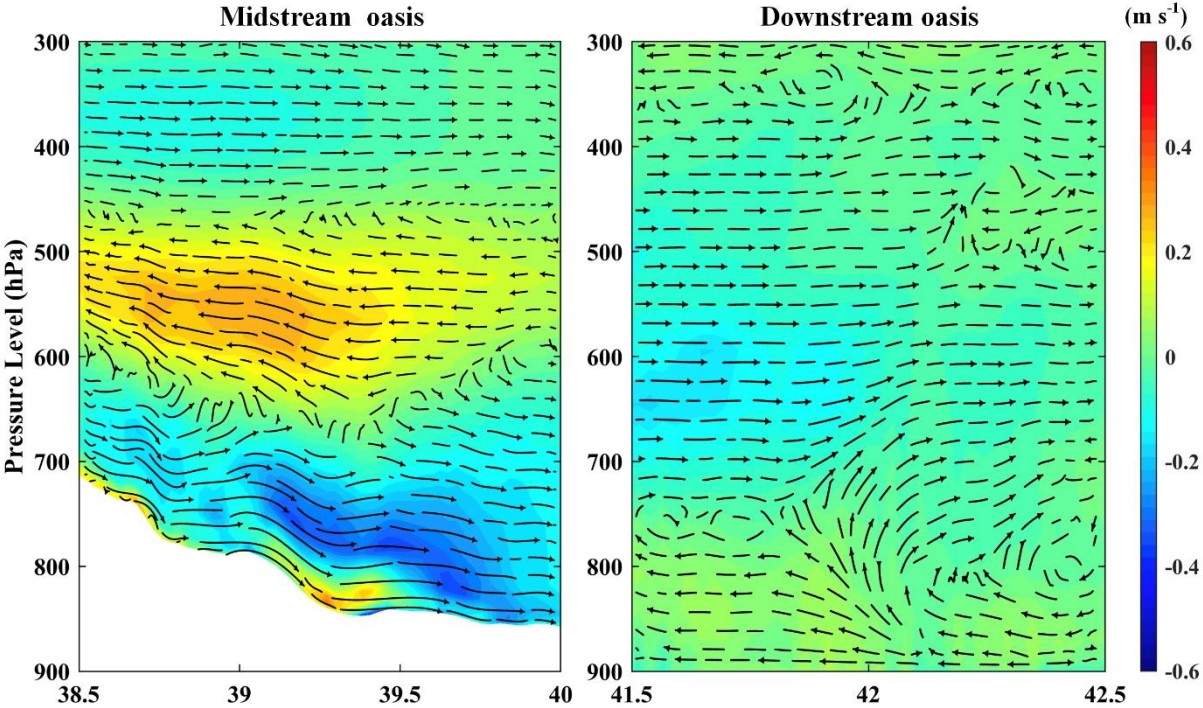

**Figure 12: Differences in the zonal mean vertical velocity and meridional circulation between the WRF (DA-ML) and WRF (OL) [WRF (DA-ML) minus WRF (OL)] in the midstream and downstream oasis. The vertical velocity in the meridional circulation is magnified 100 times for demonstration purposes.**

Figure 13 exhibits the influence of the DA-ML on precipitation in the HRB. The results show that the integrated LAI and SM led to increased precipitation in the upstream regions of the HRB and that the spatial variation in precipitation was very heterogeneous. The increase in precipitation was mainly concentrated in the southeastern part of the HRB, where it reached approximately 1.5 mm day$^{-1}$, which represented 32% of the simulated value of the WRF (OL) experiment. In contrast, precipitation increased insignificantly in the mid- and downstream oasis regions. The increased precipitation in the upstream region may have been due to the additional water vapor supply. Water vapor fluxes in the mountain areas and midstream oasis regions were enhanced by integrating the LAI and SM. Driven by background northerly winds, more water vapor fluxes from the midstream oasis region were carried to the upstream region. In addition, the integration of SM and LAI led to excessively low temperatures in the lower atmosphere and prevented convection in the irrigated regions. Therefore, rainfall may have been suppressed in the midstream region (Qian et al., 2013; Zhang et al., 2017a, 2019). The wind speed and precipitation estimates in the upstream region (around Arou station) are shown in Figure 13b and 13c. As shown, the WRF (DA-ML) enhanced the estimation of precipitation on windward slopes compared with valleys. After integrating the LAI and SM, land-atmosphere interactions are altered. The DA-ML increased the latent heat and decreases the sensible heat flux. The air masses on the valley surface were colder and wetter and had the potential for condensation and downward motion, and convection was suppressed. Simultaneously, the water vapor carried by the air masses and lifted on the sloped



surface was more likely to condense and produce precipitation. In general, the DA-ML enhanced the precipitation estimates in the upstream mountain areas, mainly on windward slopes.

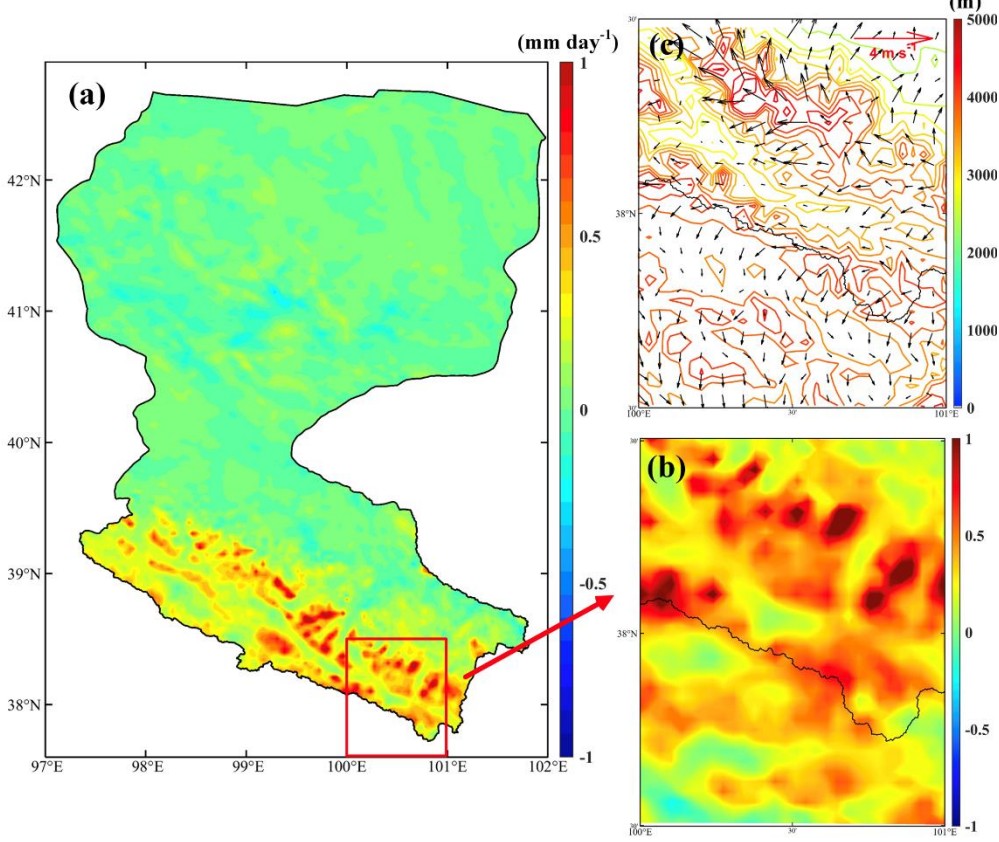

**Figure 13: (a) Average difference in precipitation between the WRF (DA-ML) and WRF (OL) [WRF (DA-ML) minus WRF (OL)] in the Heihe River Basin and (b) the upstream region (around Arou station) and (c) wind vectors at 10 m from the WRF (DA-ML) in the upstream region.**

Figure 14 compares the precipitation estimated by the WRF (OL) and WRF (DA-ML) with the precipitation observed at the nine stations. As shown, the precipitation simulated by the WRF was overestimated in the upstream regions of the HRB (except Hulugou), whereas it was more consistent in the mid- and downstream sites. After integrating the LAI and SM, more water vapor from the midstream oasis is transferred to the upstream mountains, enhancing precipitation in high-altitude regions. Weaker improvements in the simulated precipitation were observed at the mid- and downstream sites. Precipitation observations were underestimated at the upstream stations, which was largely because of uncertainty in the rain gauge measurements the caused by the obstructed airflow, saturation effects, evaporation losses, and wind-induced losses (Pan et al., 2017). In addition, snowfall observations were not included in the rain gauge measurements (Wang et al., 2017).





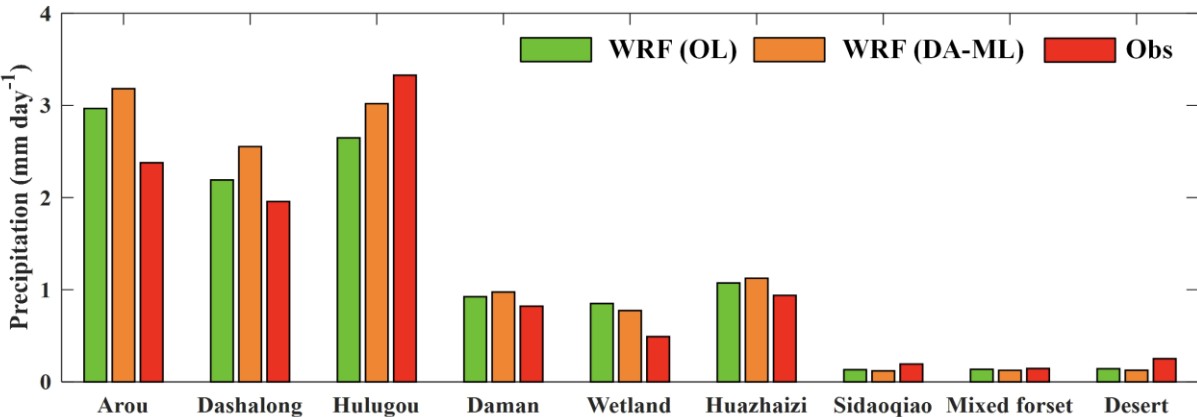


**Figure 14: Comparison of average precipitation estimates for WRF (OL) and WRF (DA-ML) with precipitation observed at nine stations for the growing season in 2015.**

The simulated daily precipitation from the WRF (OL) and WRF (DA-ML) was compared with that of the AFD and CMFD references, as shown in Figure 15. Because the water vapor from the East Asian monsoon was blocked by the
Tibetan Plateau, most of the precipitation was concentrated in the southeastern part of the Qilian Mountains. Figure 15 shows that the high precipitation zone of the HRB was mainly located in the mountainous areas below 39.5 °N due to orographic lifting and convection (Liu et al., 2017; Zhang et al., 2021b). The main precipitation events were consistent between the WRF (OL) and WRF (DA-ML). WRF (DA-ML) had higher precipitation in the southern domain because peak precipitation was enhanced and lesser precipitation events increased (red circles). Both the WRF (OL) and WRF (DA-ML)
captured the temporal and spatial variability of precipitation well and were consistent with the reference data, indicating that the 3-km high-resolution grid contains information on topography-related heterogeneity and accurately estimates the precipitation distribution. The estimated precipitation in the upstream regions of the HRB was more consistent with the AFD reference but was overestimated by approximately 0.43 mm day$^{-1}$ compared to the CMFD. The overestimation of the dynamically downscaled precipitation simulations was consistent with the simulation results from many RCMs (Pan et al.,
2021b; Xiong and Yan, 2013). The WRF model precipitation simulations exhibited a wet bias in high-altitude regions, which was partly because of the stronger precipitation-evaporation feedback (Gao et al., 2015; Yue et al., 2021). The discrepancy between the precipitation estimated by the WRF and CMFD schemes occurred because the China Meteorological Administration sites fused by the CMFD product were mainly distributed at elevations below 3500 m (He et al., 2020a). Therefore, there are some uncertainties in the precipitation simulation of the CMFD products in high-altitude mountainous
areas (Zhang et al., 2021b).



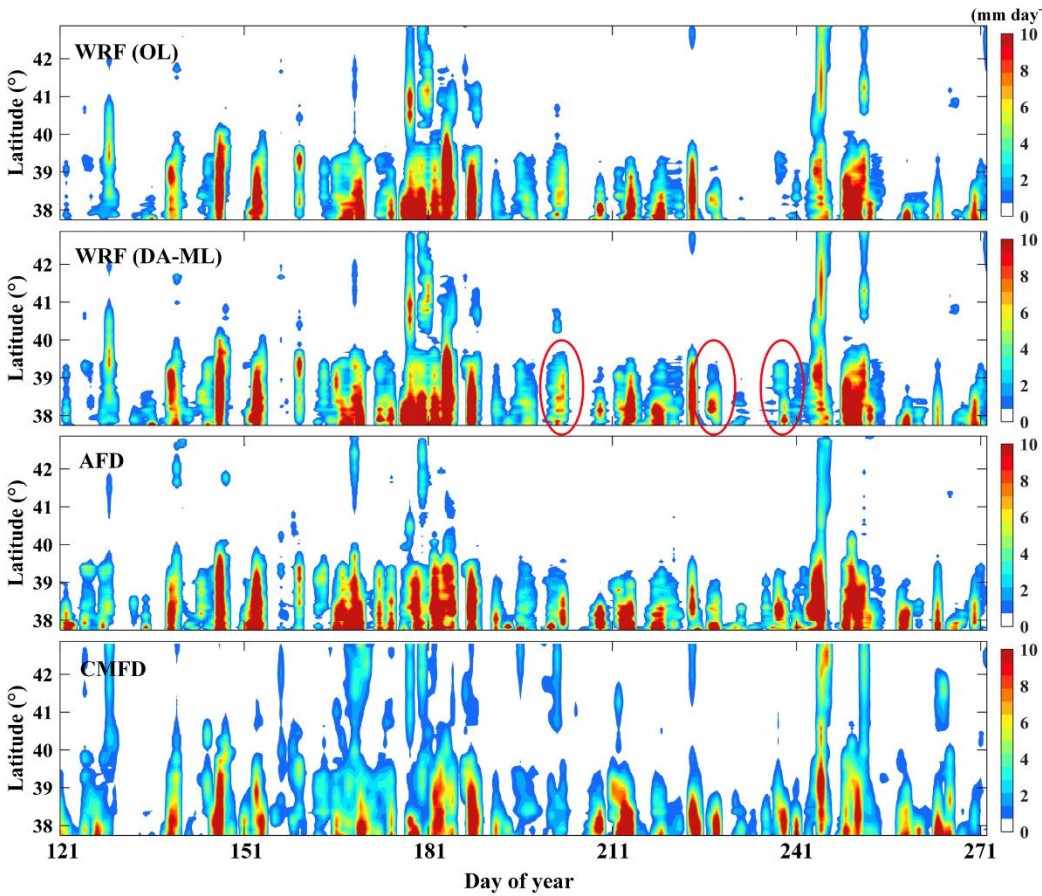

**Figure 15: Comparisons of the daily precipitation estimates from the WRF (OL) and WRF (DA-ML) with the values from the AFD and CMFD references during the growing season in 2015.**

## 5 Conclusions

In this paper, a hybrid data assimilation and machine learning framework (DA-ML approach) was proposed and implemented into the Weather Research and Forecasting (WRF) model to optimize the initialization of surface soil and vegetation variables. Remotely sensed leaf area index (LAI) and multi-source soil moisture (SM) observations (*in situ* SM profile observations and remotely sensed SM products) were integrated into the WRF model to improve the soil and vegetation characteristics. The performance of the WRF (DA-ML) framework was tested in the Heihe River Basin (HRB) in

northwestern China. The results indicated that the integration of remotely sensed LAI and multi-source SM into the WRF model improved the LAI, SM, and evapotranspiration (ET) estimates. They also indicated that the WRF (DA-ML) method improved the estimation of ET and produced a spatial distribution that was consistent with the results of ETMap. The maps of retrieved ET from the WRF (DA-ML) consistently resembled the rainfall, vegetation cover, irrigation event, and shallow groundwater table features.

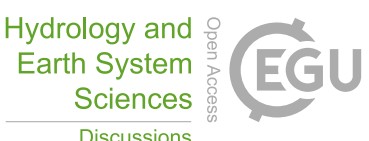

Strong wetting and cooling effects on vegetated areas were observed through the integration of LAI and SM, especially in the midstream oasis. Compared to the WRF model, the simulated seasonal cycles of air temperature and specific humidity from the WRF (DA-ML) at nine sites were closer to the station measurements. The magnitude of the surface wetting and cooling effects corresponded well with the differences in the LAI and SM estimates from the WRF (DA-ML) and WRF (OL). These results suggest that the wetting and cooling effects are related to irrigation and crop growth in the midstream oasis and shallow groundwater and riparian forest growth in the downstream oasis. This effect gradually decreased from the land surface upward and disappeared at the height of approximately 600 hPa (4000 m).

Irrigation and vegetation growth in the midstream oasis produce a wind shield effect because the stronger surface roughness. Meanwhile, the southerly airflow from the midstream oasis to the surrounding desert weakened the background north wind. Because of the wetting and cooling effects in the oasis region, wet and cold air masses are transferred to the surrounding desert by advection while dry hot air over the desert is transferred to the oasis from the upper atmosphere. Therefore, a slight warming and drying effect caused by the local meridional circulation can be observed at the height of approximately 500 hPa.

The integration of the LAI and SM leads to increased precipitation in the upstream of the HRB because the water vapor flux generated in the midstream oasis region is carried to the upstream mountains by the background northerly winds. The WRF (DA-ML) simulation captured the temporal and spatial variability of precipitation well and was consistent with the reference data. The results indicate that the 3-km high-resolution grid can consider topographic information and produce accurate precipitation distribution estimates.

**Data availability**

ERA5 data for the WRF model is freely available via the European Centre for Medium-Range Weather Forecasts (https://cds.climate.copernicus.eu/cdsapp#!/search?type=dataset). The station observations, ETMap, China Meteorological Forcing Dataset (CMFD), and atmospheric forcing data (AFD) in the HRB can be accessed from the National Tibetan Plateau Data Center (https://data.tpdc.ac.cn/en/). The assimilated LAI data are available on the Global Land Surface Satellite (GLASS) product (http://www.glass.umd.edu/). Soil moisture products can be downloaded from the Soil Moisture Active Passive (SMAP) product (https://appeears.earthdatacloud.nasa.gov/). The original WRF code can be obtained from the National Center for Atmospheric Research (NCAR) archive (https://github.com/wrf-model).

**Author contribution**

X.H. developed the model code and completed the original manuscript with support from all coauthors. Y.L., S.L., T.X., F.C., Z.L., R.L., and L.S. revised the manuscript. Z.Z. provided the original WRF code. Z.X., Z.P., and C.Z. provided the methods for processing the observational data. All authors contributed to the synthesis of the results and key conclusions.



**Competing interests**

The authors declare that they have no conflict of interest.

**Acknowledgments**

This research was partially supported by the Strategic Priority Research Program of the Chinese Academy of Sciences (XDA20100101) and the National Natural Science Foundation of China (42171315). The computation of the WRF model was supported by sources from the High Performance Computing Center of Beijing Normal University (https://gda.bnu.edu.cn/).

**Appendix A**

**Table A1: The computation details about the WRF (OL) and WRF (DA-ML).**

| Method | Computation time | Code | Processor | Parallelization |
| --- | --- | --- | --- | --- |
| WRF (OL) | 76 h | Fortran | 24-core 2.6 GHz | Intel MPI |
| WRF (DA-ML) | 112 h | Python and Fortran | Intel Xeon Gold | (100 cores) |

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
