# Peer review of "Improving regional climate simulations based on a hybrid data assimilation and machine learning method"

_Hydrology and Earth System Sciences, 2022_

## Referee Comment (RC4)

[referee-annotated manuscript omitted]

---

## Author Comment (AC1)

**Response (Referee #1 comment)**

**Ms. Ref. No.:** hess-2022-379

**Title:** Improving predictions of land-atmosphere interactions based on a hybrid data assimilation and machine learning method

This study proposed a hybrid data assimilation and machine learning framework to integrate in-situ and remotely sensed-based soil moisture observations and remotely sensed leaf area index (LAI) into the Weather Research and Forecasting (WRF) model. The ensemble Kalman filter (EnKF) approach is used to update the leaf biomass and specific leaf area by assimilating the remotely sensed LAI. A machine learning surrogate model is used to integrate soil moisture profile observations and remote sensing soil moisture product to estimate the three-layer soil moisture. In general, the hybrid framework coupled with the WRF model can improve the simulation of air temperature, specific humidity, wind speed, and precipitation, etc. in the Heihe River basin (HRB). In addition, the hybrid model can highlight the oasis-desert effect and improve the simulation of regional wind speed and precipitation. These results contribute to understanding regional climate and land-atmosphere interactions in the HRB with an advanced WRF model. The entire manuscript meets the scope of this journal. However, several points in the manuscript need to be addressed. So I suggest a minor revision is needed before publication.

We gratefully thank the reviewers for their review, which we believe has led to significant improvement on the original manuscript. The original reviewers' comments are reproduced below in black text and the corresponding response is shown in blue text.

Major comments:

1. The authors need to emphasize the advantages of the hybrid framework coupled with the WRF model compared to the previous Noah-MP model. This includes the innovative aspects of the study objectives, content, and results.

Thank you for your good comment. This study highlights the coupling of a hybrid data assimilation and machine learning approach with the WRF model and evaluates its performance in regional climate simulations. Compared with the direct assimilation of coarse-resolution remotely sensed soil moisture, this method can improve the estimation of soil moisture and ET in the heterogeneous land surface by utilizing soil moisture profile observations. Therefore, it is necessary to incorporate this hybrid approach in regional climate models to implement detailed land characterization information in basins with complex underlying surfaces and improve climate modeling.

The objective of this study is revised as: "Previous studies have also demonstrated the importance of the hybrid DA and ML method when estimating LAI, SM, and ET in typical arid/semi-arid regions of HRB. However, the advantages of improving the representation of soil and vegetation processes in affecting regional climate via the coupled DA and ML framework have not been fully exploited, especially in basins with complex underlying surfaces. Therefore, this study aims to investigate the improvement of the hybrid DA and ML framework for regional climate and land-atmosphere interactions in the HRB based on the WRF model and to further reveal its physical mechanisms."

2. Although the advantages of hybrid modeling are obvious, the authors still need to explain why ML methods were constructed to estimate soil moisture instead of directly assimilating SMAP soil moisture. In addition, the uncertainties in the estimation of soil moisture from the hybrid model need to be discussed.

Thanks for your comment. The advantages regarding the hybrid model will be added to section 3.2: "Compared with the direct assimilation of coarse-resolution remotely sensed SM, this method can improve the estimation of SM and ET on the heterogeneous land surface. This is because *in situ* SM profile observations are used to construct an ML-based surrogate model to improve SM and ET estimation on complex underlying surfaces."

The uncertainties in the estimation of soil moisture from the hybrid model will be added to section 4.1: "The results also indicate that the SM simulations from the WRF (DA-ML) model are hard to capture the observed peak values. This is because the prediction accuracy of ML methods is limited by the training data set. If the model is applied under extremely wet conditions with sparse training data, the performance of the hybrid model will decrease as the number of training samples decreases."

Minor comments:

1. How to match the spatial resolution of different datasets to the WRF system, for example, land cover data with a spatial resolution of 30 m, while WRF is set to 3 km.

Thanks for your comment. The land cover, soil texture, elevation, and GLASS LAI dataset were resampled to 3 km to be consistent with the model simulation resolution. The relevant sentences will be revised in the section 3.1.

2. The MODIS LAI is the most widely used remote sensing product. Describe why GLASS LAI can be used for assimilation instead of using other products.

Thanks for your comment. The GLASS LAI was generated using the general recurrent neural network (GRNN) approach based on Moderate Resolution Imaging Spectrometer (MODIS) and CYCLOPES LAI products. A series of validation studies have been implemented to evaluate various remotely sensed LAI products (Ma et al., 2017; Xu et al., 2018). The validation results of the global LAI products show that GLASS LAI exhibits a higher percentage of high-quality data and less uncertainty compared to other remote sensing products.

The following paragraph will be added to section 2: "The GLASS product has been demonstrated to have better accuracy than Moderate Resolution Imaging Spectroradiometer (MODIS) and Advanced Very High Resolution Radiometer (AVHRR) and provides time-space continuous LAI estimation (Xiao et al., 2014)."

3. The values of WRF (DA-ML) simulated LAI, 1.12, 1.05, 1.49, and 0.33, are obviously lower than the values drawn in Figure 2, especially at cropland. Another question is that the LAI of WRF (DA-ML) in Figure 2 is a little larger than the LAI of GLASS, not lower.

Thanks for your comment. To avoid ambiguity, the sentence is revised in section 4.1: "The simulated LAIs from WRF (OL) in the cropland, grassland, forest, and shrubland areas were 1.12, 1.05, 1.49, and 0.33 $m^2$ $m^{-2}$, respectively, all of which were lower than that of the GLASS LAI. After assimilation, the simulated bias of the LAI from WRF (DA-ML) in the HRB can be reduced from 0.94 to 0.11 $m^2$ $m^{-2}$."

4. If the horizontal coordinate in Figure 3 is Julian Day Number, its starting value should be clearly marked. Furthermore, after 200 days in the midstream, the simulation of soil moisture from the WRF (DA-ML) is hard to capture the observed peak values.

Thanks for your comment. The start horizontal coordinate values are added in Figure 3. The following sentence will be added to our manuscript: "The results also indicate that the SM simulations from the WRF (DA-ML) model are hard to capture the observed peak values. This is because the prediction accuracy of ML methods is limited by the training data set. If the model is applied under extremely wet conditions with sparse training data, the performance of the hybrid model will decrease as the number of training samples decreases."

5. Line 252: The reliability of ETMap should be described.

The following paragraph will be added to section 2: "The retrieved ET from the ETMap agrees well with the LAS observations. The multi-site averaged $R^2$, root mean square error (RMSE), and mean absolute percentage error (MAPE) values are 0.68, 0.85 mm day$^{-1}$, and 20.27%, respectively. These results confirm that the ETMap can effectively validate watershed ET simulations."

6. Line 265: In the validation work, air temperature and specific humidity simulations and observed heights need to be listed.

Thank you for your comment. The sentence is revised in section 4.3: "The monthly averaged 2 m air temperature and specific humidity from the WRF (OL), WRF (DA-ML), and corresponding observations at nine sites are shown in Figure 8 and 9."

7. Figure 6 and 7: The standard deviation of the observations is missing at Hulugou station.

Thank you for your comment. The standard deviations of the observations at Hulugou station are added to Figures 8 and 9.

8. Figures 9 and 10 show the Mean vertical profile of differences in air temperature and specific humidity between the WRF (DA-ML) and WRF (OL) during the growing season in 2015 in the midstream and downstream oasis. However, the vertical profile locations are unclear even though the rectangle has been marked in Figure 8. And I want to confirm the mean vertical profile of Figures 9 and 10 should be marked as a line or a rectangle area.

Thank you for your comment. The vertical profile simulations are averaged from the rectangular area.

9. Line 403: "The height is about 650hPa". Can hPa be converted to m?

Revised.

10. Line 417: "Driven by background northerly winds, more water vapor fluxes from the midstream oasis region were carried to the upstream region". This conclusion is hardly obtained from the Figure 13.

Thanks for your comment. The meridional circulation of the mid- and downstream oasis is added to the appendix. According to Figure A1, the airflow over the midstream oasis is mainly controlled by the background northerly wind. Therefore, the water vapor produced in the midstream oasis is carried to the upstream mountains and generates precipitation.

[Figure]

Figure A1. The zonal mean vertical velocity and meridional circulation from the WRF (DA-ML) model in the mid- and downstream oasis. The orange bar represents the oasis area.

We rephrased the sentence as "Driven by background northerly winds (Figure A1), more water vapor fluxes from the midstream oasis region were carried to the upstream region."

11. The size of the horizontal and vertical coordinates in Figure 13b and c are too small.
Thanks for your comment. The coordinates in Figure 13b and c are enlarged in the manuscript.

**References:**
Ma, H., Liu, Q., Liang, S., Xiao, Z., 2017. Simultaneous Estimation of Leaf Area Index, Fraction of Absorbed Photosynthetically Active Radiation, and Surface Albedo From Multiple-Satellite Data. IEEE Trans. Geosci. Remote Sens. 55, 4334–4354. https://doi.org/10.1109/TGRS.2017.2691542.

Xiao, Z., Liang, S., Wang, J., Chen, P., Yin, X., Zhang, L., Song, J., 2014. Use of general regression neural networks for generating the GLASS leaf area index product from time-series MODIS surface reflectance. IEEE Trans. Geosci. Remote Sens. 52(1), 209–223. https://doi.org/10.1109/TGRS.2013.2237780.

Xu, B., Li, J., Park, T., Liu, Q., Zeng, Y., Yin, G., Zhao, J., Fan, W., Yang, L., Knyazikhin, Y., Myneni, R.B., 2018. An integrated method for validating long-term leaf area index products using global networks of site-based measurements. Remote Sens. Environ. 209, 134–151. https://doi.org/10.1016/j.rse.2018.02.049.

---

## Author Comment (AC2)

**Response (Referee #2 comment)**

**Ms. Ref. No.:** hess-2022-379

**Title:** Improving predictions of land-atmosphere interactions based on a hybrid data assimilation and machine learning method

This study investigates the performance of the advanced WRF model in the HRB based on the coupled data assimilation and machine learning framework. Also, the authors assessed the impact of the hybrid framework on near-surface air conditions and land-atmosphere interactions in this region. The paper is readable and easy to follow. However, the manuscript still needs moderate revisions. Please find my comments below:

We gratefully thank the reviewers for their review, which we believe has led to significant improvement on the original manuscript. The original reviewers' comments are reproduced below in black text and the corresponding response is shown in blue text.

(1) Although the description of the manuscript is clear, I am still confused about why the authors did not directly assimilate the Soil Moisture Active Passive (SMAP) soil moisture observations. The advantages of the machine learning-based soil moisture surrogate model need to be enhanced.

Thank you for your good comment. The advantages of the SM-based surrogate model will be added to section 3.2: "Compared with the direct assimilation of coarse-resolution remotely sensed SM, this method can improve the estimation of SM and ET on the heterogeneous land surface. This is because *in situ* SM profile observations are used to construct an ML-based surrogate model to improve SM and ET estimation on complex underlying surfaces."

(2) It is not clear which soil moisture data are used for training and which data are used for validation. Independent soil moisture validation data are required.

Thanks for your comment. In this study, soil moisture profile observations from 19 automatic weather stations and SMAP SM products in the Heihe River Basin are used to train and test the soil moisture surrogate model (machine learning model). The global ten-fold testing was adopted to examine the performance of each machine learning method. This is mentioned in our manuscript (Line 186-188): "A ten-fold testing method is employed to examine the performance of each ML method. In each fold, 90% of the training samples are used to train the model, and the remaining 10% of the data is used to test the model."

In addition, soil moisture observations from the ecohydrological wireless sensor networks (WATERNET) in the up- and midstream of the HRB are used as an independent validation to evaluate the results of the WRF (DA-ML) simulations.

To address the reviewer's comment, the sentence is revised in section 2: "In this study, SM observations from the ecohydrological wireless sensor networks (WATERNET) in the up- and midstream of the HRB are used as an independent validation to evaluate the SM estimates from the WRF (DA-ML)."

(3) Line 114: The observation elements of the automatic weather stations need to be briefly described.

Thanks for your comment. The following paragraph will be added to section 2: "The AWS variables at each station include the wind speed/direction, air temperature/humidity, precipitation, air pressure, four-component radiation, photosynthetically active radiation, infrared radiation temperature, soil heat flux, and soil temperature/moisture profile (Liu et al., 2018b)."

(4) Line 132: ETMap is also uncertain and affected by assumptions. Explain why it can be used as a reference.

The following paragraph will be added to section 2: "The retrieved ET from the ETMap agrees well with the LAS observations. The multi-site averaged $R^2$, root mean square error (RMSE), and mean absolute percentage error (MAPE) values are 0.68, 0.85 mm day$^{-1}$, and 20.27%, respectively. These results confirm that the ETMap can effectively validate watershed ET simulations."

(5) Line 195: The structure diagram can provide a clear description of the coupled land-atmosphere framework.

To address the reviewer's comment, the flowchart of the hybrid model coupled with the WRF is added to the Figure 2.

[Figure]

Figure 2: (a) Details of the hybrid DA and ML method, and (b) flowchart of the coupling with the WRF model.

(6) Line 202: The details of the statistical metrics need to be listed.

To address the reviewer's comment, the following paragraph will be added to section 3.2: "The root mean square deviation (RMSD) and coefficient of determination ($R^2$) statistical metrics were used to evaluate the performance of the WRF (DA-ML) model,

$$\text{RMSD} = \sqrt{\frac{1}{n}\sum_{i=1}^{n}(P_i - O_i)^2} \tag{1}$$

$$R^2 = \frac{[\sum_{i=1}^{n}(P_i - \overline{P})(O_i - \overline{O})]^2}{\sum_{i=1}^{n}(P_i - \overline{P})^2 \sum_{i=1}^{n}(O_i - \overline{O})^2} \tag{2}$$

where $P_i$ and $O_i$ are the predicted and observed values at time step $i$, respectively. $\overline{P}$ and $\overline{O}$ represents the mean values of $P_i$ and $O_i$."

(7) In Figure 2, the WRF (DA-ML) LAI value is still higher than GLASS products in cropland. Please explain the specific reasons.
Thanks for your comment. The following paragraph will be added to section 4.1: "The results also show that the WRF (DA-ML) systematically overestimates the LAI, especially in the cropland. This is because, in addition to LAI assimilation, the integration of multi-source SM observations also affects the LAI dynamics."

(8) Line 235: "The maps of estimated LAI and SM from the DA-ML method consistently resembled the rainfall, vegetation cover, irrigation event, and shallow groundwater table features". Please explain why.
Thanks for your comment. The following paragraph will be added to section 4.1: "The precipitation in the upstream mountains, irrigation in the midstream oasis, and shallow groundwater in the downstream oasis enhance SM and provide the necessary water supply for vegetation growth (Li et al., 2022)."

(9) In Figure 5, the spatial distribution of ET estimates from the WRF (OL) and WRF (DA-ML) is more consistent in the upstream of the HRB. The authors need to explain whether the improvement of the WRF (DA-ML) is related to the original performance of the WRF (OL). Compared to ETMap, ET estimated from the WRF (OL) is underestimated in the downstream oasis, please explain more.
Thanks for your comment. The following paragraph will be added to section 4.2: "The improvement of the WRF (DA-ML) model is related to the performance of the WRF (OL). The estimation of ET in the WRF (OL) is sensitive to SM and vegetation dynamics, especially in semi-arid regions. Therefore, the WRF (DA-ML) model will produce more improvements in the mid- and downstream oasis regions compared to the WRF (OL) model."

The following paragraph will be added to section 4.2: "In addition, the higher surface heterogeneity and complex hydrological processes in the downstream oasis affect the training accuracy of the ML method, which further affects the performance of the WRF (DA-ML) model (He et al., 2022)."

**References:**
He, X., Liu, S., Xu, T., Yu, K., Gentine, P., Zhang, Z., Xu, Z., Jiao, D., Wu, D., 2022. Improving predictions of evapotranspiration by integrating multi-source observations and land surface model. Agric. Water Manag. 272, 107827. https://doi.org/10.1016/j.agwat.2022.107827.
Li, Xin, Cheng, G., Fu, B., Xia, J., Zhang, L., Yang, D., Zheng, C., Liu, S., Li, Xiubin, Song, C., Kang, S., Li, Xiaoyan, Che, T., Zheng, Y., Zhou, Y., Wang, H., Ran, Y., 2022. Linking Critical Zone With Watershed Science: The Example of the Heihe River Basin. Earth's Future, 10. https://doi.org/10.1029/2022EF002966.
Liu, S., Li, X., Xu, Z., Che, T., Xiao, Q., Ma, M., Liu, Q., Jin, R., Guo, J., Wang, L., Wang, W., Qi, Y., Li, H., Xu, T., Ran, Y., Hu, X., Shi, S., Zhu, Z., Tan, J., Zhang, Y., Ren, Z., 2018. The Heihe Integrated Observatory Network: A Basin-Scale Land Surface Processes Observatory in China. Vadose Zone J. 17, 180072. https://doi.org/10.2136/vzj2018.04.0072.

---

## Author Comment (AC3)

**Response (Referee #3 comment)**

**Ms. Ref. No.:** hess-2022-379

**Title:** Improving predictions of land-atmosphere interactions based on a hybrid data assimilation and machine learning method

This paper takes advantage of the opportunity provided by the abundance of data in the Heihe River basin to illustrate the importance of accurate soil moisture and LAI information for climate modeling in regions with highly heterogeneous land surfaces. The spatial and temporal variations of soil moisture and LAI in the WRF are realistically expressed by data assimilation and machin learning (DA+ML). After assimilating the state variables from observations or satellite remote sensing, both soil moisture content LAI values are increased, which then increases evapotranspriation in the model and futher reduces the air warming bias and dry bias in the simulation. The improved simulation shows more realistic oasis-desert boundary and the wind shield effect within the oasis. Overall, this is an excellent study in terms of capacity building that improves climate modelling through implementing detailed information of land characteristics in a basin with very complex underlying surfaces. Nevertheless, I think this paper can be organized better and some moderate revisions are required.

We gratefully thank the reviewers for their review, which we believe has led to significant improvement on the original manuscript. The original reviewers' comments are reproduced below in black text and the corresponding response is shown in blue text.

1. The scientific question to be answered is unclear. If the authors intend to answer a question of general interest, applying satellite data as an input to SM and LAI is understandable because they are globally accessible. However, the application of in situ soil moisture as an input, as done in this study, has no way to expand spatially. If the authors are trying to answer a scientific question specific to the Heihe Basin, the challenges of climate modeling in this basin should be addressed. In either case, it should be stated in the INTRODUCTION in the form of motivation.

Thank you for your good comment. This study highlights the coupling of a hybrid data assimilation and machine learning approach with the WRF model and evaluates its performance in regional climate simulations. Compared with the direct assimilation of coarse-resolution remotely sensed soil moisture, this method can improve the estimation of soil moisture and ET in the heterogeneous land surface by utilizing soil moisture profile observations. Therefore, it is necessary to incorporate this hybrid approach in regional climate models to implement detailed land characterization information in basins with complex underlying surfaces and improve climate modeling. In addition, machine learning methods have been widely used for soil moisture estimation over larger regions based on soil moisture observation networks (Li et al., 2022). Therefore, this approach can also be applied in other regions and globally.

The objective of this study is revised as: "Previous studies have also demonstrated the importance of the hybrid DA and ML method when estimating LAI, SM, and ET in typical arid/semi-arid regions of HRB. However, the advantages of improving the representation of soil and vegetation processes in affecting regional climate via the coupled DA and ML framework have not been fully exploited, especially in basins with complex underlying surfaces. Therefore, this study aims to investigate the improvement of the

[revised manuscript text omitted]

Figures 13b-c provide more details of precipitation and are retained in the manuscript. To address the reviewer's comment, the original Figure 14 and 15 was removed from the manuscript.

3. Suggest to revise the title. The work of this paper is not a prediction but a simulation; land-air interactions are not presented: it shows the response of the atmosphere to the change of the surface state, but does not present the influence of the atmosphere on the land.
Thanks for your comment. The title is revised as "Improving regional climate simulations based on a hybrid data assimilation and machine learning method."

4. L329-332: How mountain winds affect the climate in the oasis and how the cooling/wetting affects the air temperature and humidity aloft should be further clarified. Particularly, the height of 600hPa was

chosen too arbitrarily. In the oasis and desert region, the influenced height is far lower than 600hPa. Later, the authors explained the phenomenon of warmer and drier aloft through horizontal advection between oasis and desert, but I guess the enhanced subsidence over the oasis in the WRF(DA-ML) is the cause.

Thanks for your comment. It is difficult to discuss the effect of mountain winds on oases from the current results. We need additional experiments, but it is beyond the scope of the paper. Therefore, the description of the mountain winds has been removed from the manuscript.

We removed "the height of 600hPa" in the manuscript. The sentence will be revised in section 4.3: "Moreover, the wetting and cooling effects of the oasis were mainly concentrated in the boundary layer, gradually decreased from the land surface upward, and were replaced by slightly warming and drying effects. Such warming and drying effects may be related to the enhanced subsidence over the oasis. The downward motion may result in increased temperatures and bring dry air from the upper atmosphere."

5. In section 4.3, about the simulated wind speed difference between the two cases: when you update LAI, do you update the vegetation height (or roughness length and zero-plain displacement) in the WRF?

Thanks for your comment. The roughness length and zero plane displacement are constant in the WRF model based on the different land cover types. It does not change with the update of LAI.

To avoid ambiguity, we have revised the relevant sentences as: "The mean wind vectors at 10 m during the growing season from the WRF (OL) and WRF (DA-ML) in the mid- and downstream oases are shown in Figure 13. By comparing the simulated wind speeds in the oasis and the surrounding desert, we found that crops, shelterbelts, and residential areas in the midstream oasis produced a wind-shield effect. The wind speed within the oasis is less than that of the surrounding desert because the drag force of crops, shelterbelts, and residential areas reduces the wind speed and also changes the wind direction."

Minor comments:

In relevant figures, please indicate where is desert and where is oasis; otherwise, it is hard to understand what you are describing.

Thanks for your good suggestion. The indicator about the oasis area was added to Figure 9, 10, and 12.

L179: what is "the standardized soil texture"

Thank you for your comment. It refers to the normalized soil texture (ST).

L195-196: "the WRF model and DA-ML method were coupled and run dynamically and consistently through the cycles of steps one and two." What is the time interval of the cycles? This is critical information for applications.

Thanks for your comment. The following paragraph is revised in section 3.2: "Eventually, the WRF model and the DA-ML method are coupled at the daily scale through the cycles of step one and two."

L368: what you mean by "divergent wind direction". I can understand the whole sentence neither.

Thanks for your comment. We will remove the description of the mountain winds in the manuscript.

L390: In the upper atmosphere, air masses enhanced the background northerly winds (orange areas)? It is hard to understand.

Thanks for your comment. The meridional circulation of the mid- and downstream oasis is added to the appendix. According to Figure 12 and A1, the airflow from the desert to the oasis in the upper atmosphere can enhance the background northerly winds. This can be observed in the background color of Figure 12 (orange represents enhanced wind speed).

[Figure]

Figure A1. The zonal mean vertical velocity and meridional circulation from the WRF (DA-ML) model in the mid- and downstream oasis. The orange bar represents the oasis area.

The sentence is revised in section 4.4: "In the upper atmosphere, the desert to oasis air masses enhance the background northerly winds (Figure A1), which promote atmospheric water vapor transport in the HRB."

L391-395, and some similar sentences: it is not the focus of this work to study the ecological effect, which has been discussed in many early studies. Please delete.

Thanks for your comment. We deleted the relevant descriptions.

L396-397: You have not established the causality among these components.

Thanks for your comment. We deleted the relevant descriptions.

L415-418: Figure 12 shows downslope wind, so how could it transfer water vapor from the oasis upslope. There are some similar issues (e.g. L424-425). The authors must be more cautious to draw a conclusion.

Thanks for your comment. The meridional circulation of the mid- and downstream oasis is added to the appendix. As shown in Figure A1, the water vapor transport in the HRB is predominantly controlled by polar northerly winds. Driven by background northerly winds, more water vapor fluxes from the midstream oasis region were carried to the upstream region. This atmospheric water vapor transport can enhance precipitation in upstream mountainous regions (Zhang et al., 2017a).

[Figure]

Figure A1. The zonal mean vertical velocity and meridional circulation from the WRF (DA-ML) model in the mid- and downstream oasis. The orange bar represents the oasis area.

The sentence in lines 424-425 of the manuscript was deleted.

L 474: "resembled the rainfall, vegetation cover, irrigation event, and shallow groundwater table features." Is it the conclusion of this study?

Thanks for your comment. It is not the conclusion. This sentence was removed from the conclusion section.

L476: You have not presented "the simulated seasonal cycles of air temperature". Instead, you only give the seasonal mean!

The sentence is revised in section 5: "Compared to the WRF model, the seasonal mean air temperature and specific humidity simulated by the WRF (DA-ML) at the nine sites were closer to the station measurements."

---

## Author Comment (AC4)

**Response (Referee #4 comment)**

**Ms. Ref. No.:** hess-2022-379

**Title:** Improving predictions of land-atmosphere interactions based on a hybrid data assimilation and machine learning method

This is a very well-written manuscript, and this reviewer enjoy reading it through. Some major comments as below:

We gratefully thank the reviewers for their review, which we believe has led to significant improvement on the original manuscript. The original reviewers' comments are reproduced below in black text and the corresponding response is shown in blue text.

The wetting/cooling effect of the oasis is interpreted in this manuscript as WRF(DA-ML) - WRF(OL). Based on figure 9, this manuscript emphasizes that this oasis effect is related to irrigation and crop growth in the midstream. However, the wetting/cooling effect of oasis is by itself there, no matter if the DA-ML framework is applied or not. As such, this reviewer found that this manuscript is lacking of certain indices to demonstrate physically the wetting/cooling effect of the oasis (for example, one can use the difference in air temperature, and relative humidity between (above) the oasis and the surrounding areas). And then you can check how this indicator will be impacted by DA-ML (e.g. Oasis_Indictor (DA-ML) - Oasis_Indictor (OL))

Thanks for your comment. The oasis-desert interactions are caused by the different hydrothermal conditions of oases and deserts. Oasis-desert interactions lead to a series of microclimate effects, including the oasis wetting/cooling island effect and oasis wind shield effect. In this study, the proposed WRF (DA-ML) method effectively reduces air warming and drying biases in simulations, particularly in the oasis region. Therefore, this method can simulate more realistic oasis-desert boundaries, including wetting/cooling effects and wind shield effects within the oasis.

To avoid ambiguity, the relevant description is revised in section 4.3: "The above mentioned findings show that the proposed WRF (DA-ML) method exhibits wetting and cooling effects in the mid- and downstream oasis. These wetting and cooling effect reduces the air warming bias and dry bias in the simulation. Therefore, the WRF (DA-ML) simulation is much closer to the observations than the WRF (OL) simulation. Two rectangular areas (blue rectangle) were selected in Figure 10 to further analyze the effect of the DA-ML on the local climate in the mid- and downstream areas. The difference between the WRF (DA-ML) and WRF (OL) methods is used to represent the enhanced cooling and wetting effects after improving the LAI and SM simulations."

The following sentence will be added to section 4.3: "The average simulated air temperature from WRF (OL) and WRF (DA-ML) methods in the midstream oasis were 293.64 K and 291.32 K, respectively. In contrast, the near-surface air temperatures over the desert are approximately 294.13 K and 293.54 K, respectively. The difference in air temperature between the oasis and desert areas indicates that the oasis areas represent a cold and wet island compared to the surrounding desert. This difference is amplified after the implementation of the DA-ML method."

The authors state that the wetting/cooling effects of the downstream oasis are due to the shallow groundwater and riparian forest growth. This reviewer can understand that the 'riparian forest growth' can be reflected via the LAI assimilation. However, it is not explicitly clear how the shallow groundwater kicks in here. Are the authors suggesting the assimilation of root zone SM could be used to reflect the effect of shallow groundwater? If that is the case, the author should demonstrate it is indeed the case using the root zone SM, groundwater table measurements, and Noah-MP GW table simulations.

Thanks for your comment. In the downstream area of the HRB, the shallow groundwater table influences the root zone soil moisture and controls the growth of the plant communities (Xu et al., 2020; Li et al., 2022). In this study, the *in situ* SM profile observations are used as target variables to train the SM surrogate model. Therefore, compared to the WRF model, the WRF (DA-ML) method can take into account root zone soil moisture variations due to the shallow groundwater.

The spatial distribution of the root zone SM (0-100 cm) estimates from the Noah-MP and hybrid model over the HRB are shown in Figure R1. Compared to the hybrid model, Noah-MP underestimates SM in the north areas of HRB. The results indicate that soil moisture values are higher in the downstream riparian forest areas, which is due to the lateral flow of the river recharging groundwater and affecting soil moisture.

[Figure]

Figure R1. Spatial distribution of the root zone soil moisture (0-100 cm) estimates from the Noah-MP and hybrid model over the HRB in 2015.

Xu et al. (2020) analyzed the changes in groundwater table in the downstream of HRB based on the Heihe integrated observatory network. Figure R2 shows the groundwater table variations in the downstream oasis area in 2015. The groundwater was very shallow in the downstream area (approximately 1–3 m). Therefore, shallow groundwater will enhance the root zone soil moisture and accelerate vegetation transpiration.

[Figure]

Figure R2. The groundwater table variations in natural oasis in the downstream area in 2015.

The following sentence will be added to section 3.2: "Compared with the direct assimilation of coarse-resolution remotely sensed SM, this method can improve the estimation of SM and ET on the heterogeneous land surface. This is because *in situ* SM profile observations are used to construct an ML-based surrogate model to improve SM and ET estimation on complex underlying surfaces."

The following sentence will be revised in section 3.2: "The SM surrogate model can consider the effects of midstream irrigation events and downstream shallow groundwater tables on SM and improve Noah-MP ET estimates."

Although the SM surrogate model development has been published in another paper. This reviewer strongly suggested the author illustrate how these surrogate SM models were constructed with workflow/flowchart etc.

Thank you for your good comment. More details about the soil moisture surrogate model will be added to section 3.2: "In the ML part, the normalized soil texture (ST), land cover (LC), air temperature and humidity (Ta and RH), wind speed (*U*), precipitation (*P*), solar radiation (Rs), LAI, and SM observations were used to construct the SM surrogate model. ST, LC, Ta, RH, *U*, *P*, Rs, and LAI are the predictor variables. The *in situ* SM profile observations (from 19 automatic weather stations) and SMAP SM products in the HRB are used as target variables to train and test the SM surrogate model. The extreme gradient boosting (XGBoost) method was chosen in the SM surrogate model to improve multi-layer SM simulations."

To address the reviewer's comment, the flowchart of the hybrid model coupled with the WRF is shown in Figure 2.

[Figure]

Figure 2: (a) Details of the hybrid DA and ML method, and (b) flowchart of the coupling with the WRF model.

Minor comments:

L93: improving the representation of soil and vegetation processes in affecting regional climate via the coupled DA and ML ....?
Thank you for your comment. The sentence is revised as: "However, the advantages of improving the representation of soil and vegetation processes in affecting regional climate via the coupled DA and ML framework have not been fully exploited."

L165: It would be wise to illustrate how this DA-ML framework is constructed with a workflow/flowchart.
Thank you for your comment. The flowchart of the hybrid model coupled with the WRF is shown in Figure 2.

[Figure]

Figure 2: (a) Details of the hybrid DA and ML method, and (b) flowchart of the coupling with the WRF model.

L183: what are predictors? and what is the target variable? A diagram illustrating how these two surrogate models were constructed would be helpful for readers, and spare their need to read He et al. 2022 paper, in order to have this paper fully understood.

Thank you for your comment. In this study, the normalized soil texture (ST), land cover (LC), air temperature and humidity (Ta and RH), wind speed (*U*), precipitation (*P*), solar radiation (Rs), LAI, and SM observations were used to construct the SM surrogate model. ST, LC, Ta, RH, *U*, *P*, Rs, and LAI are the predictor variables. The soil moisture profile observations and SMAP SM products in the HRB are used as target variables to train and test the soil moisture surrogate model (machine learning model).

The following paragraph will be revised in section 3.2: "In the ML part, the normalized soil texture (ST), land cover (LC), air temperature and humidity (Ta and RH), wind speed (*U*), precipitation (*P*), solar radiation (Rs), LAI, and SM observations were used to construct the SM surrogate model. ST, LC, Ta, RH, *U*, *P*, Rs, and LAI are the predictor variables. The *in situ* SM profile observations (from 19 automatic weather stations) and SMAP SM products in the HRB are used as target variables to train and test the SM surrogate model."

In addition, more details about the SM surrogate model are illustrated in Figure 2.

[Figure]

Figure 2: (a) Details of the hybrid DA and ML method, and (b) flowchart of the coupling with the WRF model.

L193: The LAI is updated via Noah-MP, and the SM is updated via the surrogate model?

Thank you for your comment. Yes, the LAI is updated via Noah-MP, and the SM is updated via the surrogate model. The sentence will be added to section 3.2: "In this step, the LAI is updated by the DA method, and the SM is updated via the ML-based surrogate model."

L360: but this is kept unchanged in either WRF(OL) or WRF (DA-ML), right?

Thank you for your comment. The bulk transfer coefficient is changed after the implementation of the DA-ML method. The mean bulk transfer coefficient shown in Figure 11 is to compare the surface roughness in the oasis with that of the surrounding desert.

The sentence is revised in section 4.4: "In Figure 13, the bulk transfer coefficient ($C_h$) from the WRF (DA-ML) was used to compare the surface roughness in the oasis with that of the surrounding desert."

**References:**

Li, Xin, Cheng, G., Fu, B., Xia, J., Zhang, L., Yang, D., Zheng, C., Liu, S., Li, Xiubin, Song, C., Kang, S., Li, Xiaoyan, Che, T., Zheng, Y., Zhou, Y., Wang, H., Ran, Y., 2022. Linking Critical Zone With Watershed Science: The Example of the Heihe River Basin. Earth's Future 10. https://doi.org/10.1029/2022EF002966

Xu, Z., Liu, S., Zhu, Z., Zhou, J., Shi, W., Xu, T., Yang, X., Zhang, Y., He, X., 2020. Exploring evapotranspiration changes in a typical endorheic basin through the integrated observatory network. Agricultural and Forest Meteorology 290, 108010. https://doi.org/10.1016/j.agrformet.2020.108010